# Ensemble forecast of an index of the Madden Julian Oscillation using a stochastic weather generator based on circulation analogs

Meriem Krouma [1,2], Riccardo Silini [3], and Pascal Yiou [2]

[1]ARIA Technologies, 8 Rue de la Ferme, 92100 Boulogne-Billancourt, France.
[2]Laboratoire des Sciences du Climat et de l'Environnement, UMR 8212 CEA-CNRS-UVSQ, IPSL & Université Paris-Saclay, 91191 Gif-sur-Yvette, France.
[3]Departament de Fisica, Universitat Politècnica de Catalunya, Edifici Gaia, Rambla Sant Nebridi 22, 08222 Terrassa, Barcelona, Spain.

**Correspondence:** Meriem Krouma (meriem.krouma@lsce.ipsl.fr)

**Abstract.**

The Madden-Julian Oscillation (MJO) is one of the main sources of sub-seasonal atmospheric predictability in the Tropical region. The MJO affects precipitation over highly populated areas, especially around Southern India. Therefore, predicting its phase and intensity is important as it has a high societal impact. Indices of the MJO can be derived from the first principal
components of zonal wind and outgoing longwave radiation (OLR) in the Tropics (RMM1 and RMM2 indices). The amplitude and phase of the MJO are derived from those indices. Our goal is to forecast these two indices on a sub-seasonal timescale. This study aims to provide an ensemble forecast of MJO indices from analogs of the atmospheric circulation, computed from the geopotential at 500 hPa (Z500) by using a stochastic weather generator (SWG). We generate an ensemble of 100 members for the MJO amplitude for sub-seasonal lead times (from 2 to 4 weeks). Then we evaluate the skill of the ensemble forecast
and the ensemble mean using probabilistic scores and deterministic skill scores. According to score-based criteria, we find that a reasonable forecast of the MJO index could be achieved within 40-day lead times for the different seasons. We compare our SWG forecast with other forecasts of the MJO. The comparison shows that the SWG forecast has skill compared to ECMWF forecast for lead time above 20 days and better skill compared to machine learning forecasts for small lead times.

## 1 Introduction

Forecasting the Madden Julian Oscillation (MJO) is a crucial scientific endeavor as the MJO represents one of the most important sources of subseasonal predictability in the tropics. The Madden Julian oscillation controls tropical convection, with a life cycle going from 30 to 60 days (Lin et al., 2008). It is characterized by a dominant eastward propagation over the tropical Indo-Pacific basin in particular during the boreal winter. The MJO affects the Indian, Australian monsoons (Zhang, 2013), and West African monsoon (Barlow et al., 2016). It was shown that it affects precipitation in East Asia (Zhang et al., 2013) and
North America (Becker et al., 2011). The MJO affects the global weather as it impacts the tropics as well as the extratropics due to the atmospheric teleconnections (Zhang, 2013; Cassou, 2008).

The improvement of the forecast skill of the MJO is subject of several studies. Numerical models have shown an ability to forecast the MJO index (Kim et al., 2018). However, the forecast of the MJO is sensitive to the quality of the initial conditions (Zhang, 2013; Straub, 2013). This motivates probabilistic forecasts to overcome the chaotic nature of climate variability (Sivillo et al., 1997; Palmer, 2000). Indeed, ensemble forecasts have shown improvements over deterministic forecasts for weather and climatic variables on short and long term (Yiou and Déandréis, 2019; Hersbach et al., 2020). One of the advantages of ensemble forecasts is that they provide information about the forecast uncertainties, which deterministic forecasts cannot provide. In addition, the use of ensemble means has shown better forecast results than the individual ensemble members in previous works (Toth and Kalnay, 1997; Grimit and Mass, 2002; Xiang et al., 2015).

Statistical models, such as stochastic weather generators (SWG), have been used for this purposes. SWGs are designed to mimic the behavior of climate variables (Ailliot et al., 2015). They have been used to forecast various weather and climatic variables such as temperature (Yiou and Déandréis, 2019), precipitation (Krouma et al., 2021) and the North Atlantic oscillation (NAO) (Yiou and Déandréis, 2019). One of the benefits of using stochastic weather generators is that they have a low computing cost compared to numerical models. Combining stochastic weather generators with analogs of the atmospheric circulation is an efficient approach to generate ensemble weather forecasts with consistent atmospheric patterns (Yiou and Déandréis, 2019; Krouma et al., 2021; Blanchet et al., 2018).

Analogs of circulation were designed to provide forecast assuming that similar situations in the atmospheric circulation could lead to similar local weather conditions (Lorenz, 1969). Recent studies have evaluated the potential of analogs to forecast the probability distribution of climate variables: Yiou and Déandréis (2019) simulated large ensemble members of temperature using random sampling of atmospheric circulation analogs; Atencia and Zawadzki (2014) used analogs of precipitation to forecast precipitation.

The goal of this study is to forecast a daily MJO index for a subseasonal lead time ($\approx 2 - 4$ weeks) with a SWG based on analogs of the atmospheric circulation, described in Sec. 3.2. The SWG approach was evaluated in previous studies by Yiou and Déandréis (2019) and Krouma et al. (2021) for European temperature and precipitation. The SWG was able to forecast the temperature within 40 days and the precipitation within 20 days with reasonable skill scores in western Europe (Krouma et al., 2021; Yiou and Déandréis, 2019). In this paper, we adjust the parameters of the SWG in order to forecast the MJO indices. More precisely, our goals are (i) to forecast the MJO amplitude (directly from the amplitude, and using the MJO indices), and (ii) to evaluate the ability of our SWG model to forecast active events of the MJO for the following weeks. We will evaluate the sensitivity of the SWG approach on the forecast with different seasons and compare the forecast skill using SWG to other forecast approaches.

The paper is divided as follows: Section 2 shows the data used for running our forecast. Section 3 explains the methodology: circulation analogs, stochastic weather generator and the verification metrics that we used to evaluate the SWG forecast. Section 4 explains the experimental setup. Section 5 details results of simulations and the evaluation of the ensemble forecast. Section 6 is devoted to the comparison of the SWG forecast with the literature. Section 7 contains the main conclusions of the analyses.

 ## 2  Data

The MJO has been described by various indices, that are obtained from different atmospheric variables (Stachnik and Chrisler, 2020). Wheeler and Hendon (2004) defined an MJO index from two so-called Real-time Multivariate MJO series (RMM). RMM1 and RMM2 represent respectively the first and second principal components of the empirical orthogonal functions (EOFs) resulting from the combination of daily fields of the satellite-observed outgoing longwave radiation (OLR), and the

zonal wind at 250 hPa and 850 hPa latitudinally averaged between 15°N and 15°S (Rashid et al., 2011). The EOFs are computed from daily normalized fields after applying a filter to remove the long timescale variability (annual mean and the first three harmonics of the seasonal cycle), the previous 120 days of anomaly fields and the El Niño signal as described by Wheeler and Hendon (2004). Lim et al. (2018) and Ventrice et al. (2013) proposed other indices proposed of the MJO. The main difference between the indices consists in the input fields and the computation of the index. For instance, Ventrice et al. (2013) replace

OLR with 200hPa velocity potential and Lim et al. (2018) do not remove an El Niño signal.

The RMM1 and RMM2 allow to compute the amplitude and the phase of the MJO (Wheeler and Hendon, 2004). For this paper, we selected the RMM-based MJO index. One of the reasons is that it is often used for MJO forecast (e.g. Kim et al., 2018; Rashid et al., 2011; Silini et al., 2021).

To simplify notations in the equations, we will note $R_1 = \text{RMM1}$ and $R_2 = \text{RMM2}$. The amplitude ($A$) and phase ($\phi$) are

defined as follows:

$$A(t) = \sqrt{R_1(t)^2 + R_2(t)^2}, \tag{1}$$

and

$$\phi(t) = \tan^{-1} \frac{R_2(t)}{R_1(t)}. \tag{2}$$

The amplitude and the phase describe respectively the evolution of the MJO and its position along the equator. The amplitude

is related to the intensity of the MJO activity. There are different classifications related to the intensity of the active MJO events (Lafleur et al., 2015). In this paper, we consider that there is an MJO event when $A(t) \geq 1$ (Lafleur et al., 2015). The phase $\phi$ is decomposed into eight areas known as centers of convection of the MJO over the equator, starting from the Indian Ocean through the maritime continent to the western Pacific Ocean. This leads to a discretization $\hat{\phi}$ of phase $\phi$ into those eight identified areas (Lafleur et al., 2015). For each day $t$, we consider the amplitude $A(t)$, which can be above 1 (active MJO) or

below 1, and the phase $\hat{\phi} \in \{1, \ldots, 8\}$. The amplitude and the phase are usually represented in a phase-space diagram (Lafleur et al., 2015), called the Wheeler-Hendon phase diagram. An example of Wheeler-Hendon phase diagram is shown in Figure 1.

We obtained daily time series of RMMs, amplitude ($A$) and phase ($\hat{\phi}$) from January 1979 to December 2020 over the region covering 15°N − 15°S, from IRI (2022) (Wheeler and Hendon, 2004). In this paper, we aim at forecasting RMM variations.

We used the geopotential at 500 hPa (Z500), 300 hPa (Z300) and Outgoing Longwave Radiation (OLR) daily data to compute

the analogs. The data are available from 1948 to 2020 with a horizontal resolution of $2.5° \times 2.5°$. The data were downloaded from the National Centers for Environmental Prediction (NCEP, Kistler et al. (2001)).

In this paper, we predict the daily amplitude $A$ and phase $\phi$ of the MJO, from the daily analogs of Z500, Z300 and OLR.

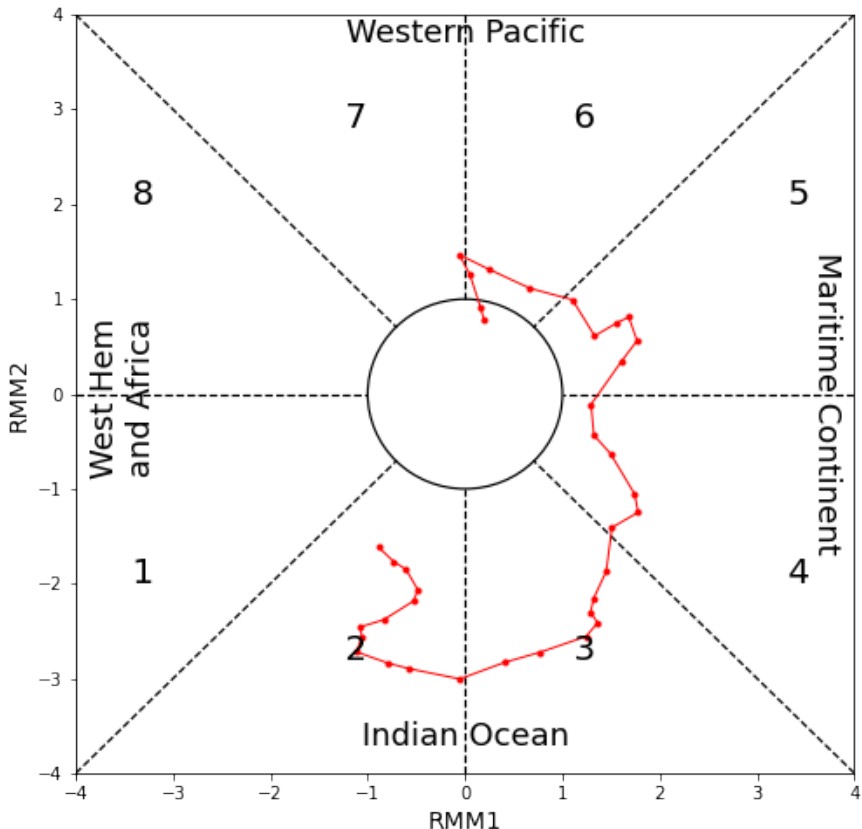

**Figure 1.** Wheeler-Hendon phase diagram of the MJO event for the period between 1986/03/03 and 1986/04/09, for observations. The Diagram shows the 8 areas of activity of MJO starting from the Indian Ocean.

## 3 Methodology

### 3.1 Analog computation

We start by building a database of analogs. For a day $t$, we define analogs as dates $t'$ within 30 calendar days of $t$ that have a similar Z500 (or Z300 or OLR) configuration as $t$. We look for analogs in different years from $t$. We quantify the similarity between daily Z500 maps using the Euclidean distance. The analogs are computed from daily data using a moving time window of $\Delta = 30$ days. This duration $\Delta$ corresponds to the life cycle of the MJO. Then, we keep the 20 best analogs. We define "best

analog" as dates which have the minimum Euclidean distance between $t$ and $t'$. The use of the Euclidean distance and the
number of the analogs were explored and justified in previous studies (Krouma et al., 2021; Platzer et al., 2021).

Hence the distance that is optimized to find analogs of the $Z500(x,t)$ field is:

$$D(t,t') = \left[ \sum_x \left( \sum_{i=0}^{\tau} |Z500(x,t+i) - Z500(x,t'+i)|^2 \right) \right]^{\frac{1}{2}}, \tag{3}$$

where $x$ is a spatial index, $\tau$ is a time window size (e.g. $\tau = 3$ days).

We compute separately analogs of Z500, Z300 and OLR following the same procedure over the Indian Ocean as represented
in Figure 2. We adjusted the parameters of computation of the analogs mainly the search window of the analogs and the
geographical domain. We considered different geographical regions to search for analogs. We computed analogs over the Indian
ocean, the Indian-Pacific ocean and the Indian-maritime ocean for verification purposes (Annex B1). This lead to consider an
optimal region for the analog search outlined in Figure 2.

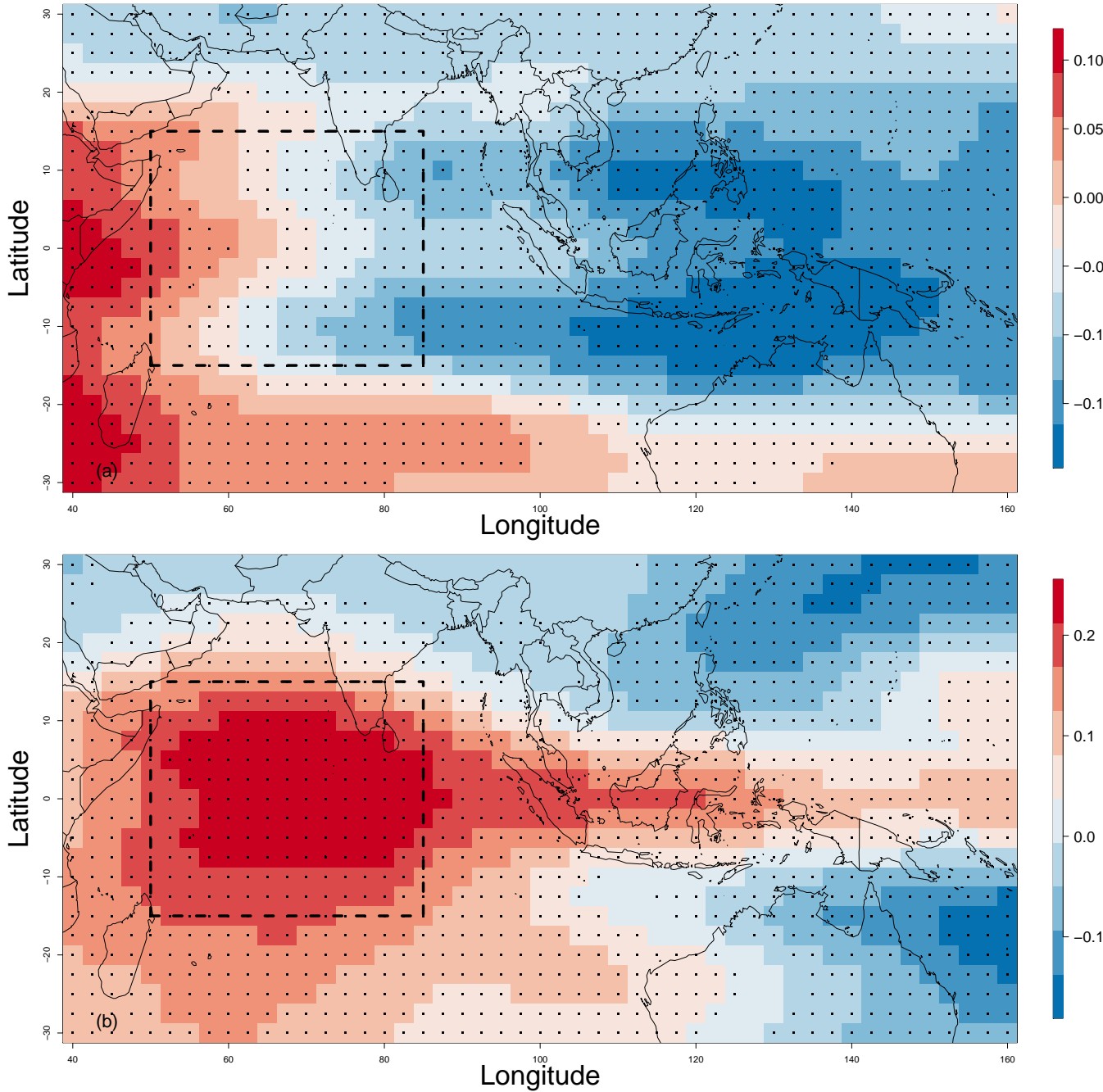

**Figure 2.** The optimal domain of computation of analogs, we computed analogs over the Indian ocean, in the geographic areas indicated by the black dash rectangle with coordinates (50E – 85E; 15S – 15N). The figure shows the temporal correlation between Z500, RMM1 (panel a) and RMM2 (panel b) for the whole studied period from 1979 to 2020. The correlation is weak but it is still significant with p-values $\leq 0.05$ that we indicated by black dots over each grid of the considered domain (including the optimal region used to compute analogs).

## 3.2 Configuration of the stochastic weather generator

The stochastic weather generator (SWG) aims to generate ensembles of random trajectories that yield physically consistent features. Our SWG is based on circulation analogs that are computed in advance with the procedure described in Section 3.1 (Yiou, 2014; Krouma et al., 2021). We produce an ensemble hindcast forecast with the circulation analog SWG with the following procedure (see Figure 3 for a summary).

For a given day $t_0$ in year $y_0$, we generate a set of $S = 100$ simulations until a time $t_0 + T$, where $T$ is the lead time, which goes from 3 to 90 days. We start at day $t_0$ and randomly select an analog (out of $K = 20$) of day $t_0 + 1$. The random selection of analogs of day $t_0 + 1$ among $K$ analogs is performed with a weight $w_k$ that is computed as the products of two weights $w_k^c$ and $w_k^\phi$ defined by the following rules:

1. weights $w_k^c$ are inversely proportional to the calendar difference between $t_0$ and analog dates, to ensure that time goes "forward". If $\delta_k$ is the difference of calendar days between $t_0 + 1$ and $t_k$, where $t_k$ is the date of the $k$th analog of $t_0 + 1$, then the calendar day sampling weight $w_k^c$ is proportional to $\exp(-|\delta_k|)$.

2. weights $w_k^\phi$ are the difference between the phase at $t_0$ and analog dates. Indeed, we give more weight to analogs that are in the same phase. If $\delta'_k$ is the difference between $\hat{\phi}(t_0 + 1)$ and the discrete phase $\hat{\phi}_k$ of $t_k$, then the phase weight $w_k^\phi$ is proportional to $\exp(-|\delta'_k|)$.

Then we set $w_k = 0$ when the analog year is $y_0$. Indeed, excluding analog selection in year $y_0$, ensures that we do not use information from the $T$ days that follow $t_0$. Then $w_k = w_k^c \times w_k^\phi$ and the values of $w_k$ are normalized so that their sum is 1. Rule 1 is similar to the SWG used by Krouma et al. (2021). Rule 2 adds a constraint to ensure phase consistency across analogs.

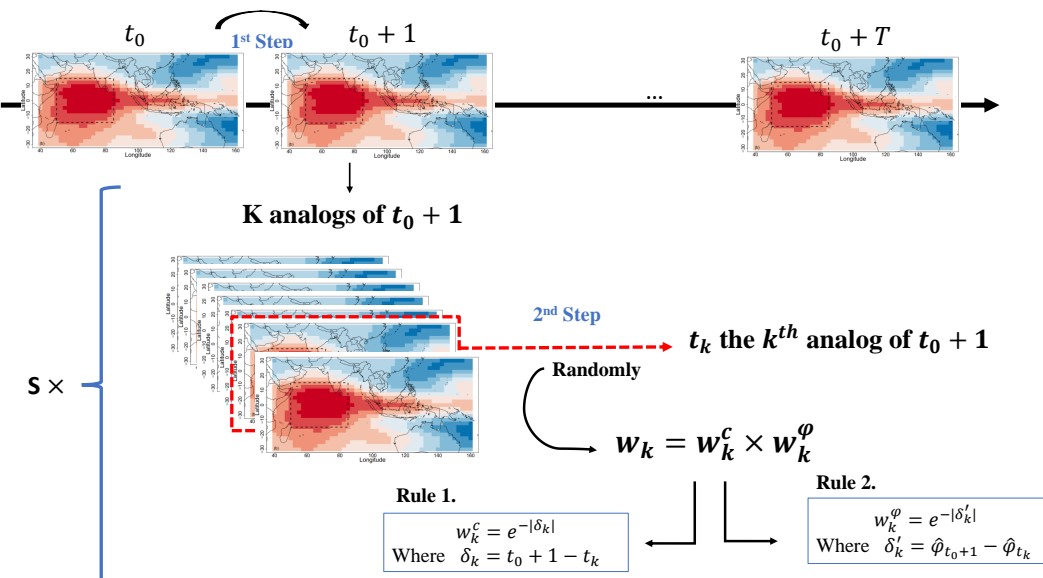

**Figure 3.** Illustration the SWG process. The 1st step goes from one day to the next day. The 2nd Step explains how we randomly select a $k$th analog with respect to weight $w_k$.

We then replace $t_0$ with $t_k$ the selected analog of $t_0 + 1$ and repeat the operation $T$ times. Hence we obtain a hindcast trajectory between $t_0$ and $t_0 + T$. This operation of trajectory simulation from $t_0$ to $t_0 + T$ is repeated $S = 100$ times. The daily MJO ($A(t)$ or RMMs) of each trajectory is time-averaged between $t_0$ and $t_0 + T$. Hence, we obtain an ensemble of $S = 100$ forecasts of the average MJO ($A(t)$ or RMMs) for day $t_0$ and lead time $T$. Then $t_0$ is shifted by $\Delta t \geq 1$ days, and the ensemble simulation procedure is repeated. This provides a set of ensemble forecasts with analogs.

To evaluate our forecasts, the predictions made with the SWG are compared to the persistence and climatological forecasts. The persistence forecast consists of using the average value between $t_0 - T$ and $t_0$ for a given year. The climatological forecast takes the climatological mean between $t_0$ and $t_0 + T$. The persistence and climatological forecasts are randomized by adding a small Gaussian noise, whose standard deviation is estimated by bootstrapping over $T$ long intervals. We thus generate sets of persistence forecasts and climatological forecasts that are consistent with the observations (Yiou and Déandréis, 2019).

### 3.3 Forecast verification metrics

We assess the skill of the SWG to forecast the $A(t)$ and the RMMs using two approaches. We start by evaluating the performance of the SWG to forecast $A(t)$. For that, we use probabilistic scores (Zamo and Naveau, 2018; Hersbach, 2000; Marshall et al., 2016), like the Continuous Rank Probability Score (CRPS) for each lead time $T$. The CRPS is a quadratic measure of the difference between the forecast cumulative distribution function and the empirical cumulative distribution function of the observation (Zamo and Naveau, 2018). The CRPS is defined by:

$$CRPS(P,x_a) = \int\limits_{-\infty}^{+\infty} (P(x) - \mathcal{H}(x - x_a))^2 dx, \tag{4}$$

where $x_a$ is the observed $RMM^{obs}$ or $A(t)^{obs}$, $P$ is the cumulative distribution function of $x$ of the ensemble forecast and $\mathcal{H}$ represents the Heaviside function, ($\mathcal{H}(y) = 1$ if $y \geq 0$, and $\mathcal{H}(y) = 0$ otherwise). A perfect forecast yields a CRPS value equal to 0.

As the CRPS value depends on the unit of the variable to be predicted, it is useful to normalize it with the CRPS value of a reference forecast, which can be obtained by a persistence or a climatology hypothesis. The CRPSS is defined as a percentage of improvement over such a reference forecast (Hersbach, 2000). We compute the CRPSS using as a reference the climatology and the persistence.

$$CRPSS = 1 - \frac{\overline{CRPS}}{\overline{CRPS_{ref}}}, \tag{5}$$

where $\overline{CRPS}$ is the average of the $CRPS$ of the SWG forecast and $\overline{CRPS_{ref}}$ is the average of the $CRPS$ of the reference (either climatology or persistence).

The CRPSS values vary between $-\infty$ and 1. The forecast has improvement over the reference when the CRPSS value is above 0.

We also computed the rank (temporal) correlation between the observations and the median of the 100 simulations (Scaife et al., 2014).

A robust forecast requires a good discrimination skill. A discrimination skill represents the ability to distinguish events from non-events. We measure the skill of the SWG in discriminating between situations leading to the occurrence of an MJO event (active MJO) and those leading to the non-occurrence of the event (inactive MJO). To do so, we use the relative operating characteristic (ROC) score. The ROC is used for binary events (Fawcett, 2006). Since we have a probabilistic forecast, we can use a threshold value of 1 to construct a classifier for the binary event of MJO from the feature $A(t)$:

  – If $A(t) \geq 1$ we predict a positive outcome (active MJO),

  – if $A(t) < 1$ we predict a negative outcome (inactive MJO).

The ROC curve is a plot of the success rate versus the false alarm rate (Verde, 2006). The ROC curve could be also a plot of the *sensitivity* versus the *specificity* (Fawcett, 2006). The sensitivity (true positive rate) is the probability of an active MJO event, assuming that $A(t) \geq 1$ is really observed. The specificity (true negative rate), refers to the probability of an inactive MJO event, as long as we have $A(t) \leq 1$. Moreover, the sensitivity is a measure of the ability of the prediction to identify true positives and the specificity is a measure of the ability to identify true negatives. Both quantities describe the accuracy of a prediction that signals the presence or absence of a MJO event (Fawcett, 2006). Therefore, we define the relationship between sensitivity and specificity as follow:

- $specificity = 1 - sensitivity$ means that we have a poor prediction because the rate of true negative and the false alarm rate are the same,

- $specificity > 1 - sensitivity$ means that we have a good prediction.

Another performance measurement that we can infer from the ROC curve is the Area Under the Curve (AUC). The AUC explains how much the forecast model is able to distinguish between binary classes. The AUC is the area in the ROC curve between sensitivity and the false positive rate computed as follow:

$$AUC = \int_0^1 S(x)dx \tag{6}$$

Where $S$ is the sensitivity and $x$ is the false positive rate.

An increase in AUC indicates an improvement in discriminatory abilities of the model at predicting a negative outcome as a negative outcome and a positive outcome as a positive outcome. An AUC of 0.5 is non-informative.

Finally, we evaluate the ensemble mean forecast of $RMM1$ and $RMM2$ using the usual scalar metrics for MJO forecast (Rashid et al., 2011; Silini et al., 2021; Kim et al., 2018). We computed the bivariate anomaly correlation coefficient (COR) and the bivariate root mean square error (RMSE) between the forecasted RMMs ($R_i^{pred}$) and the observed RMMs ($R_i^{obs}$) as follow:

$$COR(T) = \frac{\sum_{t=1}^{t=N}[R_1^{obs}(t)R_1^{pred}(t,T) + R_2^{obs}(t)R_2^{pred}(t,T)]}{\sqrt{\sum_{t=1}^{t=N}[R_1^{obs}(t)^2 + R_2^{obs}(t)^2]}\sqrt{\sum_{t=1}^{N}[R_1^{pred}(t,T)^2 + R_2^{pred}(t,T)^2]}}, \tag{7}$$

$$RMSE(T) = \sqrt{\frac{\sum_{t=1}^{t=N}[|R_1^{obs}(t) - R_1^{pred}(t,T)|^2 + |R_2^{obs}(t) - R_2^{pred}(t,T)|^2]}{N}}, \tag{8}$$

where $t$ is the time, $T$ is the lead time of the forecast, $N$ is the length of the time series ($N \sim 10^4$). We interpret the values of COR and RMSE using thresholds fixed by previous studies to define the forecast skill of the SWG. The forecast has skill when the COR value is larger than $0.5$ and the RMSE value is lower than $\sqrt{2}$. Rashid et al. (2011) explain that for a climatological forecast, $RMSE = \sqrt{2}$ because the standard deviation of the observed RMM indices is 1. Hence, forecasts are considered to be skilful for $RMSE < \sqrt{2}$ (i.e., they have lower RSME than a climatological forecast). We will use those threshold in our analyses.

We compare the RMSE to the ensemble spread in order to evaluate the forecast accuracy. The ensemble spread measures the difference between the members of the ensemble forecast. The ensemble spread $ES$ is obtained by the root mean square difference between the ensemble members and the ensemble mean defined as follow:

$$ES = \sqrt{\frac{\sum_{n=1}^{S}(A_n - \overline{A})^2}{S}}, \tag{9}$$

where $S$ is the size of the ensemble members, $A_n$ is the amplitude of the $n$th ensemble member of the forecast and $\hat{A}$ is the ensemble average of $A_n$ over the $S$ members.

We compute the average amplitude error ($E_A$) and the average phase error ($E_\phi$) for the different lead times $T$. They allow to evaluate the quality of the forecast. The average amplitude error ($E_A$) is defined as follow:

$$E_{A(T)} = \frac{1}{N} \sum_{t=1}^{t=N} [A_{pred}(t,T) - A_{obs}(t)], \tag{10}$$

The value of $E_{A(T)}$ indicates how fast the forecast system loses the amplitude of the MJO signal. If a positive value indicates an overestimation of the amplitude in predictions compared to the observation. A negative value indicates an underestimated amplitude. Rashid et al. (2011) define the average phase error ($E_\phi$) as:

$$E_{\phi(T)} = \frac{1}{N} \sum_{t=1}^{t=N} \tan^{-1} \frac{R_1^{obs}(t) R_2^{pred}(t,T) - R_2^{obs}(t) R_1^{pred}(t,T)}{R_1^{obs}(t) R_1^{pred}(t,T) + R_2^{obs}(t) R_2^{pred}(t,T)}. \tag{11}$$

This formulation stems from the ratio of the cross product (numerator) and dot product (denominator) of the vectors of forecast ($R_1^{pred}, R_2^{pred}$) and observations ($R_1^{obs}, R_2^{obs}$). Eq. (11) is equivalent to the average phase angle difference between the prediction and observations, with a positive angle indicating the forecast leads the observations (Rashid et al., 2011). The negative (positive) value of $E_{\phi(T)}$ indicates a slower (faster) propagation of the phase in predictions compared to the observations.

## 4   Forecast Protocol

We explore the skill of a SWG to forecast the $A(t)$ and the $RMMs$ ($R_1$ and $R_2$) using analogs of the atmospheric circulation. We generate separately an ensemble of 100 members of the $A(t)$ of the MJO and RMMs using the same approach. The goal is to have a probabilistic forecast of the $A(t)$ for a subseasonal lead time $T$ ($\approx$ 2 to 4 weeks). As input to the SWG, we are using analogs of the atmospheric circulations. We computed analogs separately from Z500, Z300, the wind at 250 hPa and 850 hPa and the OLR. We choose to keep analogs from the geopotential at height 500 hPa instead of the other atmospheric fields. We explain our choice in section 5.

Then, we adjusted the geographical region and the window search of analogs Annexe-figure B1. Indeed, the forecast skill of the MJO depends on the geographical region. We choose to compute the analogs over the Indian Ocean with coordinates (50°E – 85°E; 15°S –15°N). We argued our choice by the fact that (i) the Indian Ocean corresponds to the first phase of the MJO in the phase-space diagram, where the MJO starts. (ii) Different models found good results by initiating their forecast in this region (Kim et al., 2018), (iii) and based on the experiment analyses that we made over different geographical regions Annexes-Figure B2. We explained that in the Annexe B.

We search for analogs within 30 calendar days. This duration corresponds to the life cycle of the MJO. In addition, we adjust the SWG in order to select analogs from the same phase, as described in Section 3.2.

To evaluate the skill score of our forecasts, we used two approaches. We used the probabilistic scores such as CRPS, correlation and ROC score (section 3.3) to evaluate the ensemble forecast of the amplitude. Then, we evaluate the ensembles

mean of $RMM1$ and $RMM2$. For that, we used scalar metrics such as the COR and the RMSE (section 3.3), as they are
225 commonly used to evaluate MJO forecast (Rashid et al., 2011; Lim et al., 2018).

## 5  Results

We show results of the forecast of $A(t)$ and $RMMs$ ($R_1$ and $R_2$) from the analogs of Z500 over the Indian Ocean with a time
of search of 30 days. As explained in section 4, we explored the potential of other atmospheric circulations (wind at 250 hPa
and 850 hPa, OLR and Z300) to forecast the MJO amplitude. The forecast skill with analogs of OLR and the zonal wind at the
230 upper and lower troposphere (250 and 850 hPa) was not that satisfying compared to the forecast skill using analogs of Z500 or
Z300 Figure 4. Indeed, the wind at 250 hPa, 850 hPa, and the ORL do not improve the bivariate correlation and RMSE forecast
skill of the MJO index for a longer lead time (above 20 days) over Z500 or Z300 Figure 4, despite the fact they are the driver
of the MJO. This could be explained by different reasons.

The first reason is related to the composition of the RMMs index. Indeed, the OLR is used as a proxy for organized moist
convection (Kim et al., 2018). However, the fractional contribution of the convection to the variance of RMMs is considerably
lower than the fraction of the zonal wind fields (Kim et al., 2018; Straub, 2013). The second reason is that the MJO predictability
can be improved by including atmospheric and oceanic processes (Pegion and Kirtman, 2008). According to some theories that
explain the MJO, the geopotential and the moisture are considered as a driver of precipitation and convection(Zhang et al.,
2020). For instance, in the gravity-wave theory for MJO (Yang and Ingersoll, 2013), the convection and precipitation are
240 triggered by a specific geopotential threshold.

Another reason is related to our forecast approach. The composites of OLR and wind speed highly depend on the phase of
the MJO (Kim et al., 2018). As our analog approach is constrained by choice of a geographical region, it misses the spatio-
temporal variability of OLR and wind speed during the MJO. We computed analogs from other regions (Annexes: Figure B1).
However, we obtain better forecast scores by focusing on the "small" area represented by a dashed rectangle (Annexes - Figure
B1). This is explained by the higher quality of analogs. Indeed, choosing a "large" region to compute analogs yields rather
large distances or low correlations for analogs. This implies that the analog SWG gets lower skill scores because the analogs
are not very informative. The OLR or zonal wind analogs were computed on the optimal window obtained for Z500 or Z300
as mentioned in Figure 2 which is not appropriate for OLR or wind speed, as reflected by Kim et al. (2018). Therefore, we
find lower COR and RMSE scores compared to the forecast using Z300 and Z500. This is a potential feature of analogs. The
250 analog geometry needs to be imposed a priori in a rather simplistic way, which does not follow the spatio-temporal features of
the MJO, which are known independently.

We tested the forecast of $A(t)$ and RMMs using analogs of Z300. We get a satisfactory forecast skill (i.e. with $COR > 0.5$ and
$RMSE < \sqrt{2}$) up to $T = 60$ days. However, we notice that the forecast skill scores based on analogs of Z500 are higher for
small lead times (up to 30 days). This is explained by the fact that Z300 analogs are close to where the MJO takes place, even
if this does not lead to significant improvement over Z500 analog skill scores. Therefore the geopotential heights, although
less physically and dynamically relevant for the MJO, are more appropriate predictors from the statistical and mathematical

constraints of the analog-based method. The results of the forecast with analogs of Z300 can be found in the Annexe A where we compared the performance of the SWG forecast based on the analogs of Z500 and Z300 for different seasons Annexes-Figures A2 and A1. For those reasons, we decided to keep the results of the forecast for $A(t)$ and RMMs with analogs of Z500.

This analysis highlights the capacity of Z500 to catch the variability of the MJO.

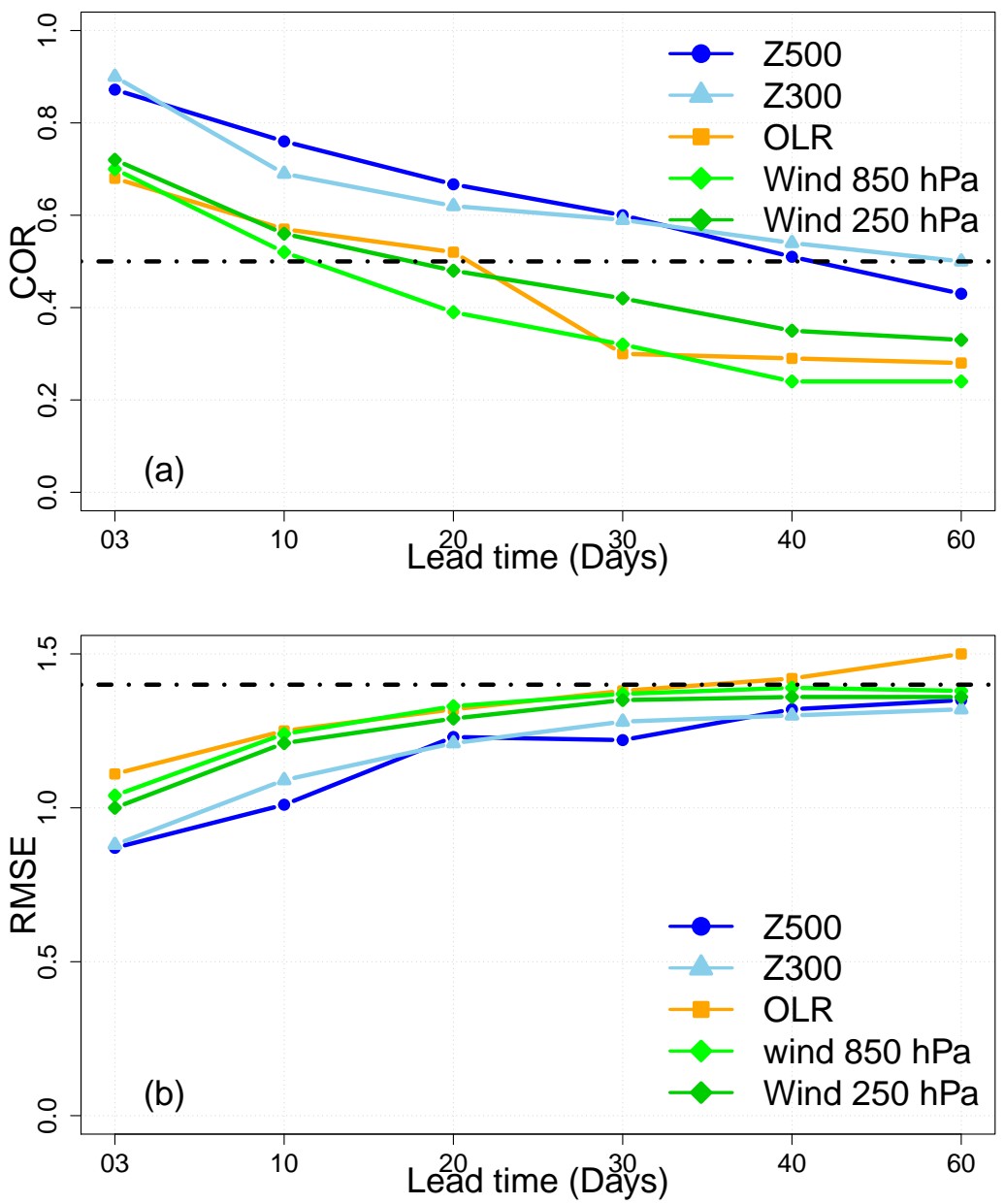

**Figure 4.** COR (panel a) and RMSE (panel b) values for different lead time of forecast from 3 days to 60 days over the period from 1979 to 2020 for the SWG forecast using analogs of OLR, zonal wind speed at 250 hPa and 850hPa, Z300 hPa and Z500 hPa.

As an illustration, we show the time series of the simulations and observations of the MJO amplitude for 1986. This year yields an unusually large period of RMM amplitude above 1, suggesting an important MJO activity. Figure 5 shows the mean of the 100 simulations and the observations for lead times of 3, 5 and 10 days for the whole year. We find that there is a strong

correlation between observed and simulated $A(t)$ for the different lead times represented. Moreover, the SWG was able to distinguish between the active MJO days ($A(t) \geq 1$) and inactive MJO ($A(t) < 1$). The same figures for the forecast with the SWG based on analogs of OLR and Z300 are provided in the Annexe A.

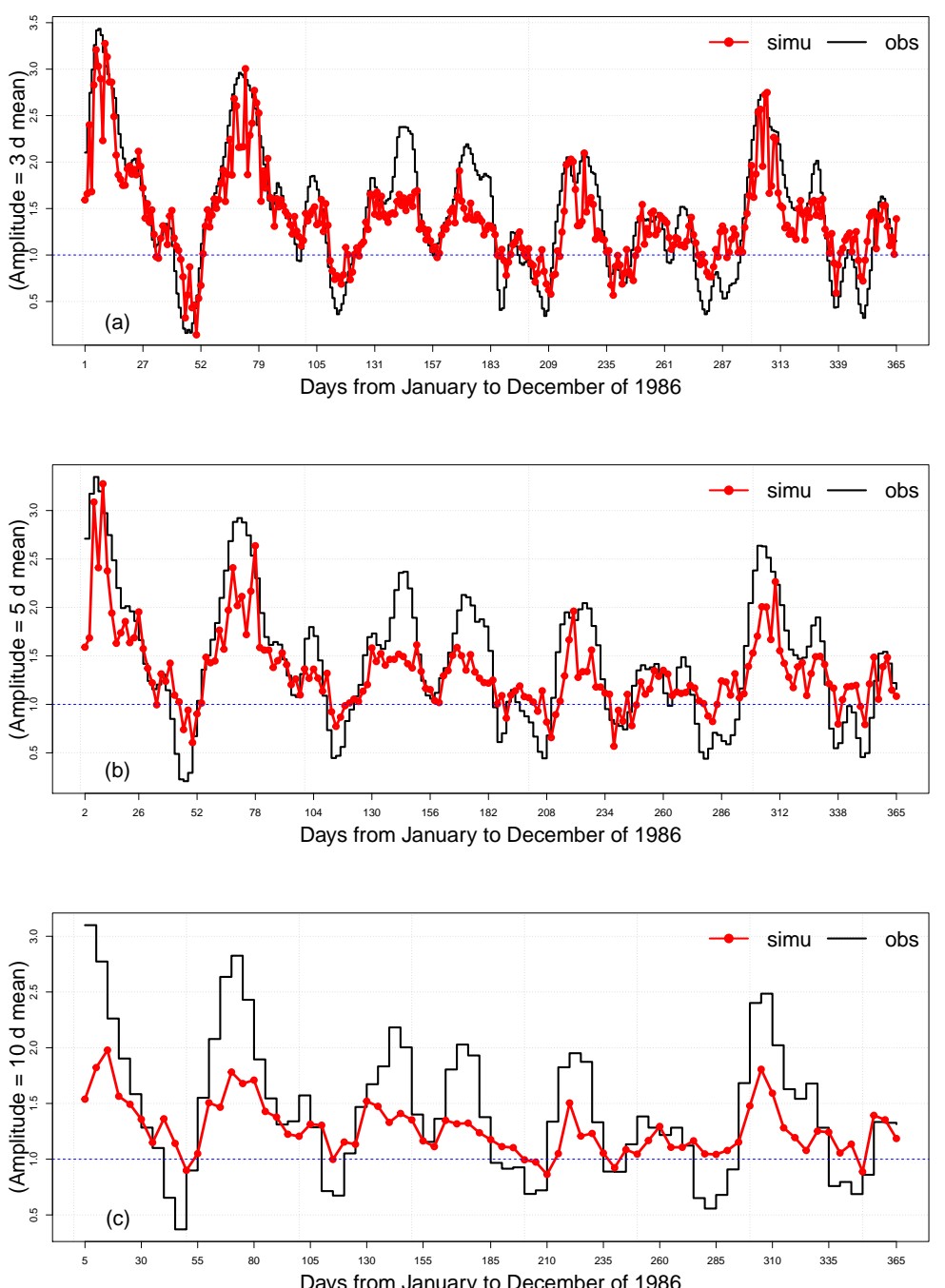

**Figure 5.** Time series of observations and simulations of the MJO Amplitude for lead time of 3, 5 and 10 days, respectively (a), (b) and (c), for the year 1986. The red line represents the mean of the 100 simulations, the black line represents the observations, and the blue line indicates the threshold of the MJO activity (below 1: inactive; above 1: active)

## 5.1 Evaluation of the Ensemble forecast of the MJO Amplitude

We evaluate the forecast of amplitude $A(t)$ using the probabilistic skill scores (CRPSS, ROC and correlation) defined in Section 3.3. We are considering the average of the skill scores up to each lead time $T$. In Figure 6, we show the CRPSS and the correlation for DJF (December, January and February) and JJA (June, July and August) for different lead times $T$ going from 3 to 40 days.

The CRPSS was computed using as a reference the forecast made from climatology and persistence. We notice that the CRPSS vs. persistence reference is decreasing with time. It has higher values for $T = 3, 5, 10$ days. We notice that when the lead time is larger than $T = 15$ days, CRPSS values become stable for both seasons. However, the CRPSS vs. climatology is increasing with lead times. We notice that for small lead times ($T \leq 15$ days), the SWG forecast is doing better than the persistence, while for big lead times $T \geq 15$ days, the SWG forecast is doing better than the climatology. We can say that the forecast has positive improvement compared to climatology and persistence for DJF and JJA for all the studied lead times. We see that correlation is mostly decreasing with lead times. The highest correlation is related to small lead times ($T \leq 15$ days).

We used the ROC diagram to determine the discrimination between active and inactive events of the MJO. We associated 1 to active MJO event and zero to the inactive events. In Figure 7, we show the ROC diagram for the different lead times $T$ from 3 to 40 days. Analyzing the AUC, shown in Table 1, we find that until 40 days, the SWG is able to separate non-events (inactive MJO) from events as the AUC values are between 0.88 and 0.61. It is still significant as it is over the diagonal (random forecast). We notice that the sensitivity value is $0.9$ for 3 days, and it decreases with lead time to reach $0.7$ by 40 days. We also find that the specificity and sensitivity are equal for small lead times. However, the specificity remains above $\approx 0.5$ for $T = 40$ days. This value of specificity is still higher than ($1 - sensitivity = 0.2$). This indicates that the forecast has skill to distinguish between MJO events until 40 days ahead.

**Table 1.** Area under ROC curve (AUC) for the different lead times $T$ from 3 to 40 days

| $T$ | 3 days | 5 days | 10 days | 20 days | 30 days | 40 days |
|-----|--------|--------|---------|---------|---------|---------|
| AUC | 0.88 | 0.83 | 0.74 | 0.66 | 0.62 | 0.61 |

Using three probabilistic metrics (CRPSS, correlation and ROC), we show that the SWG is able to skillfully forecast the MJO amplitude from analogs of Z500. The CRPSS shows a positive improvement of the forecast until 40 days. However, the correlation is significant until 20 days. By using the ROC curve and the discrimination skill, we show that the forecast still has skill until 40 days.

The difference between the lead times that we found using the CRPSS, correlation and the ROC result from the difference between the skill scores. In fact, the CRPS is used for different categories of events, while the ROC is used for binary events, which is more suitable with our case of study.

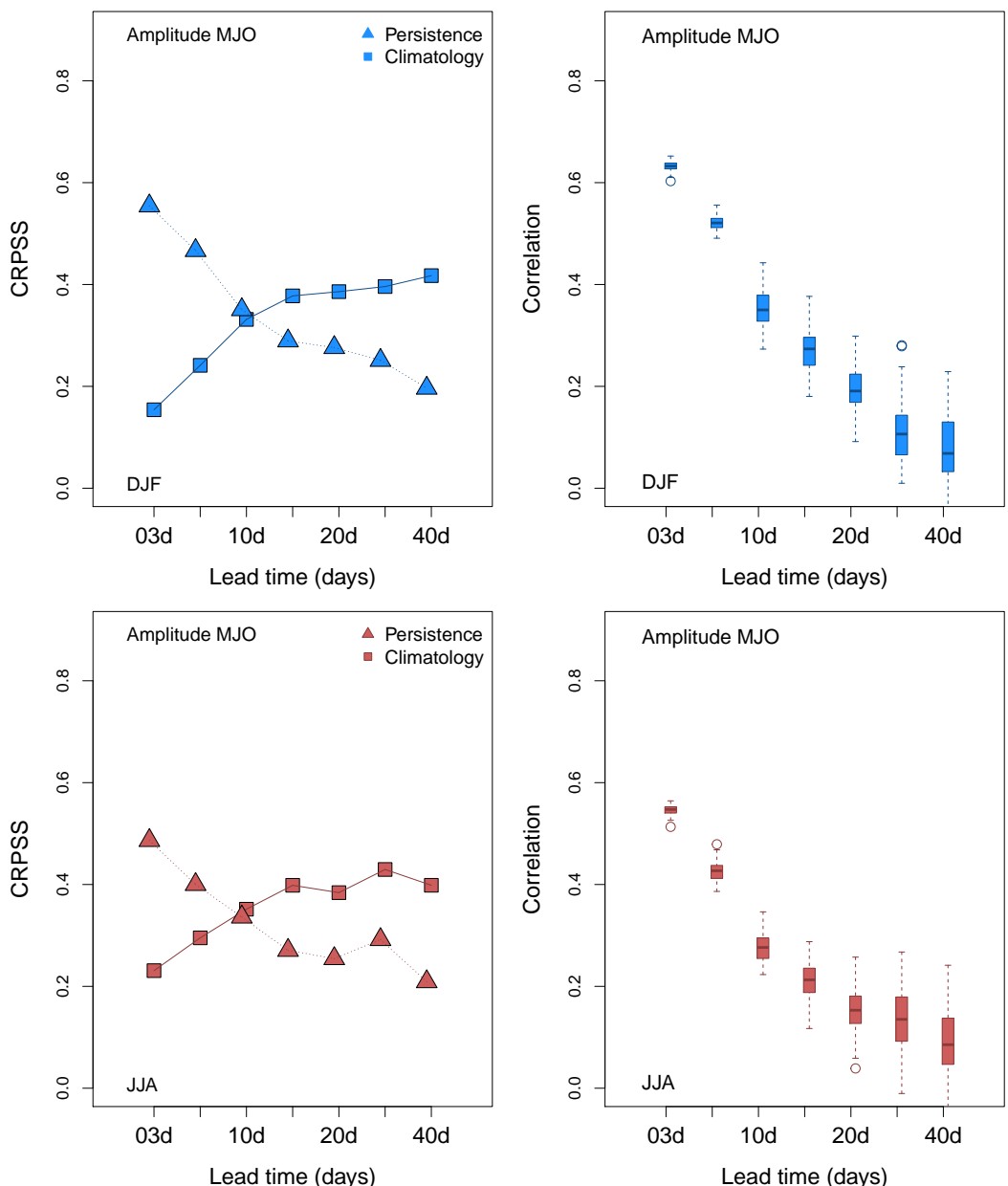

**Figure 6.** Skill scores for the MJO Amplitude for lead times going from 3 to 40 days for DJF (blue) and JJA (red) for analogs computed from Z500. Squares indicate CRPSS where the Persistence is the reference, triangles indicates CRPSS where the climatology is the reference, and box-plots indicates the probability distribution of correlation between observation and the median of 100 simulations for the period from 1979 to 2020.

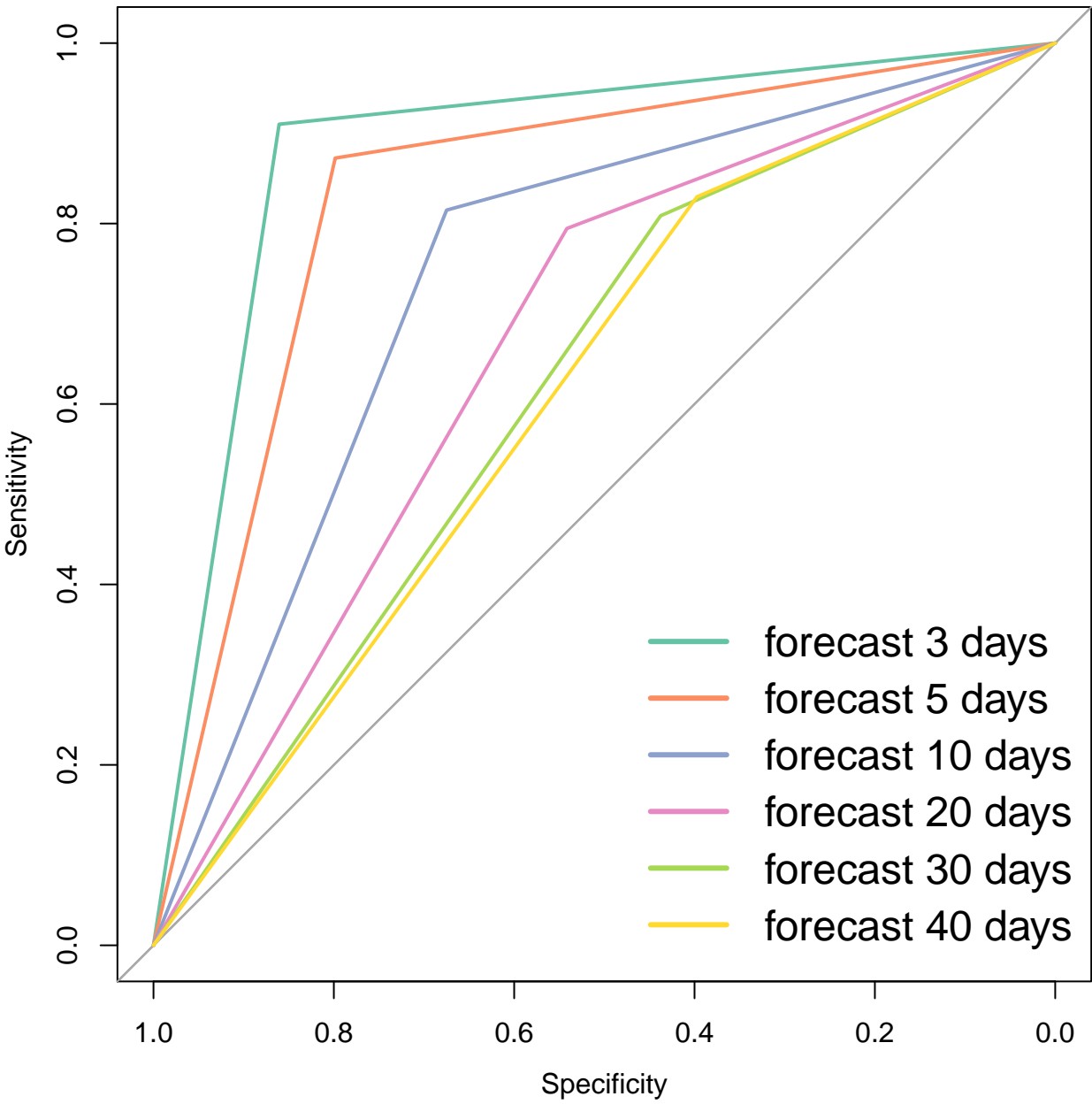

**Figure 7.** ROC curve for all lead times. The plot represents the sensitivity versus the specificity. The diagonal line represents the random classifier obtained when the forecast has no skill. If the ROC curve is below the diagonal line, then the forecast has a poor skill, otherwise it has a good skill, i.e. the forecast has the potential to distinguish between success and false alarm.

## 5.2  Evaluation of the Ensemble mean forecast of RMMs

In this part, we evaluate the performance of the SWG to forecast the RMMs ($R_1$ and $R_2$). We simulated $R_1$ and $R_2$ using the SWG and analogs of Z500. Then we used the ensemble mean of $R_1$ and $R_2$ to compute the verification metrics mainly the COR and RMSE (Rashid et al., 2011; Kim et al., 2018; Silini et al., 2021), as shown in Figure 8. We looked at COR and RMSE averaged up to each lead time $T$. Respecting the threshold $0.5$ for the COR and $\sqrt{2}$ for RMSE, we found that the forecast has skill until $T = 40$ days. We have to mention that $T$ of 60 and 90 days were used for verification purposes.

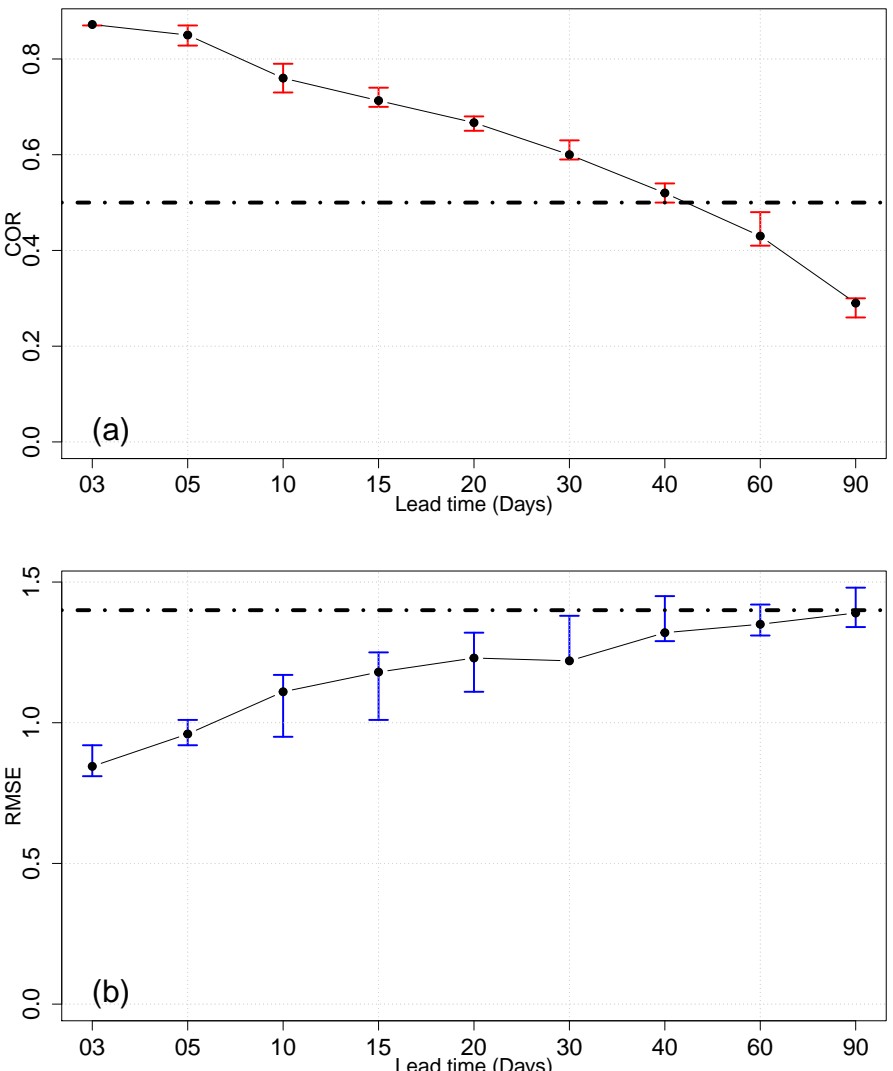

**Figure 8.** The COR and RMSE, respectively (a) and (b), for the different lead time of forecast from 3 days to 90 days over the period from 1979 to 2020. Confidence intervals are obtained with a bootstrap with 1000 samples.

In order to verify the forecast skill, we computed the ensemble spread and we compared it to the RMSE values for the different lead times going from 3 to 40 days (Figure 9). We found that the difference between the ensemble spread and the RMSE is increasing with lead time. The RMSE is becoming larger with lead time, which indicates that the distance between the observations and simulations is increasing. In addition, the ensemble spread is decreasing, which indicates that the uncertainties are increasing with time. This was verified by computing the bias of the forecast where we could find that it is increasing with

lead time. The bias represents the average bias of RMM1 and RMM2. It was computed between the ensemble mean of the RMMs and the observations of RMMs.

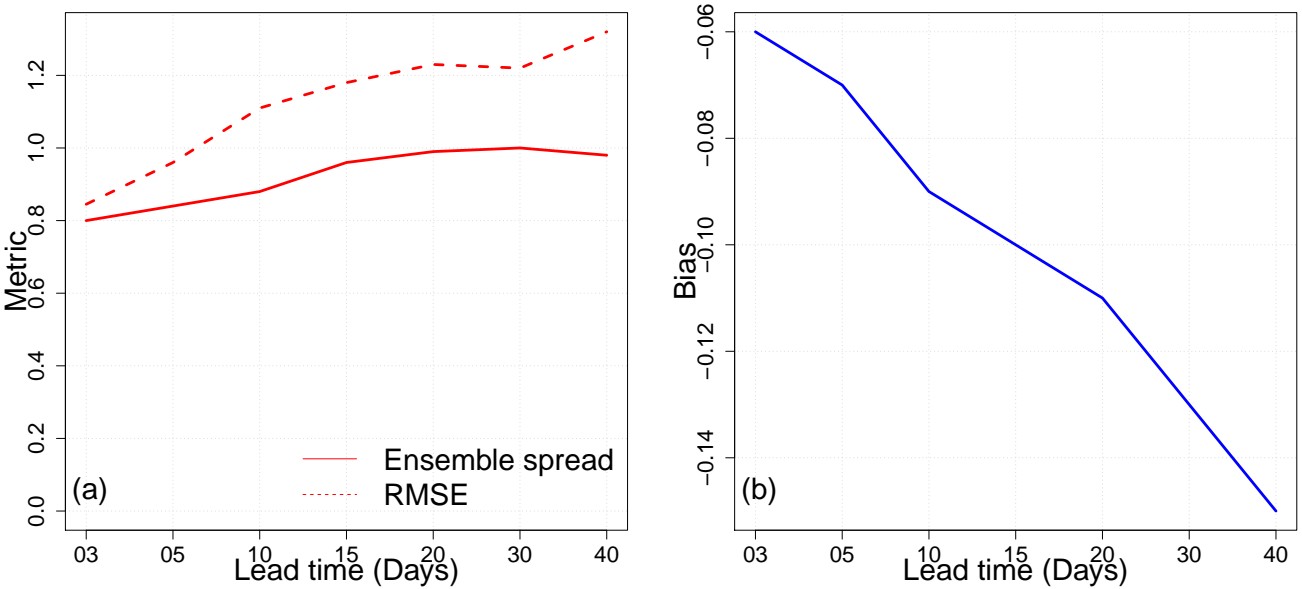

**Figure 9.** (a) Comparison between the ensemble spread and the RMSE. We notice that the difference is small for short lead times ($\leq 15$ days). "Metric" in the vertical axes refers to Ensemble spread and RMSE. (b) The bias between the simulations and the observations for the lead time going from 3 to 40 days.

  We explored the sensitivity of the forecast to seasons as shown in Figure 10. We found that the forecast for DJF and MAM (March, April, May) has a good skill (i.e., with RMSE lower than $\sqrt{2}$) within 30 days. However, for SON (September, October and November) and JJA, a similar forecast skill was obtained for a lead time of 40 days. The DJF and MAM seasons show the

310 largest RMSE values. This implies that the ensemble forecast in DJFM yield a larger range of values than in SON and JJA, even if the observations and simulations are well correlated. The highest correlation in DJF and MAM could be explained by the fact that the MJO is more active in the boreal winter (DJFM). However, the RMSE values in JJA are more consistent as they represent low distance between simulations and observations. Indeed, even if the MJO events tend to be more intense in DJFM, the amplitude is underestimated. The assessment of the ensemble mean forecast of RMM1 and RMM2 showed that the

315 forecast has skill until 40 days. However, it is sensitive to seasons and this is consistent with the previous studies of Wheeler

and Hendon (2004); Rashid et al. (2011); Wu et al. (2016b). Indeed, we found that the SWG forecast of RMM1 and RMM2 has skill, with respect to the thresholds of COR and RMSE, within 40 days for summer (JJA) and 30 days for winter (DJF).

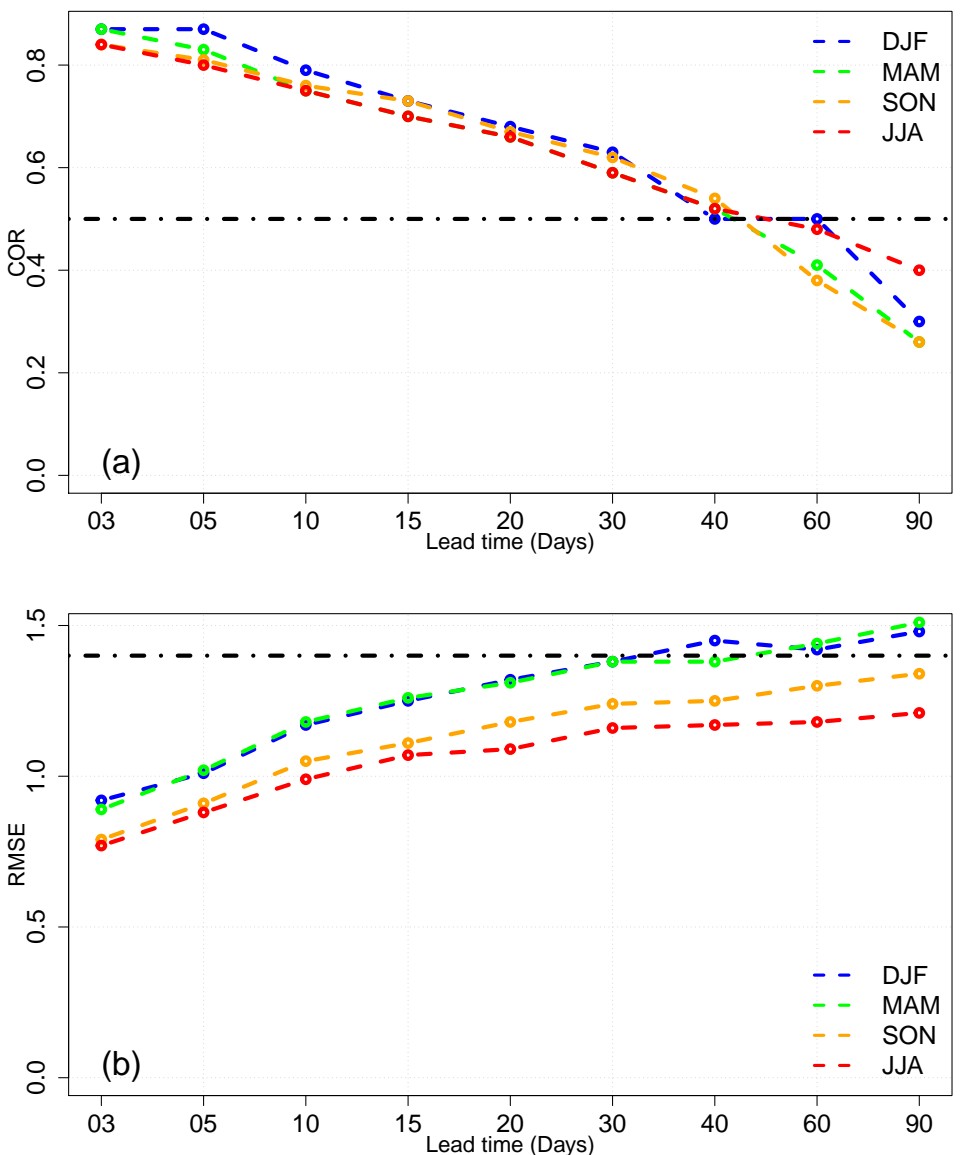

**Figure 10.** The COR and RMSE, respectively (a) and (b), for the different lead time of forecast from 3 days to 90 days over the period from 1979 to 2020 for the different seasons DJF, JJA, MAM and SON.

We also computed the amplitude and phase errors (Figure 11). We found that the $E_{A(t)}$ is negative for all lead times. That indicates a weak amplitude in predictions compared to the observations. The $E_{\phi(t)}$ is positive until 30 days, which indicates

fast propagation of the phase in predictions compared to the observations. Then it becomes negative, which means that the phase is slower. We notice that the phase is well predicted while the amplitude is underestimated Figure 11. This is consistent with previous studies (Silini et al., 2021; Rashid et al., 2011).

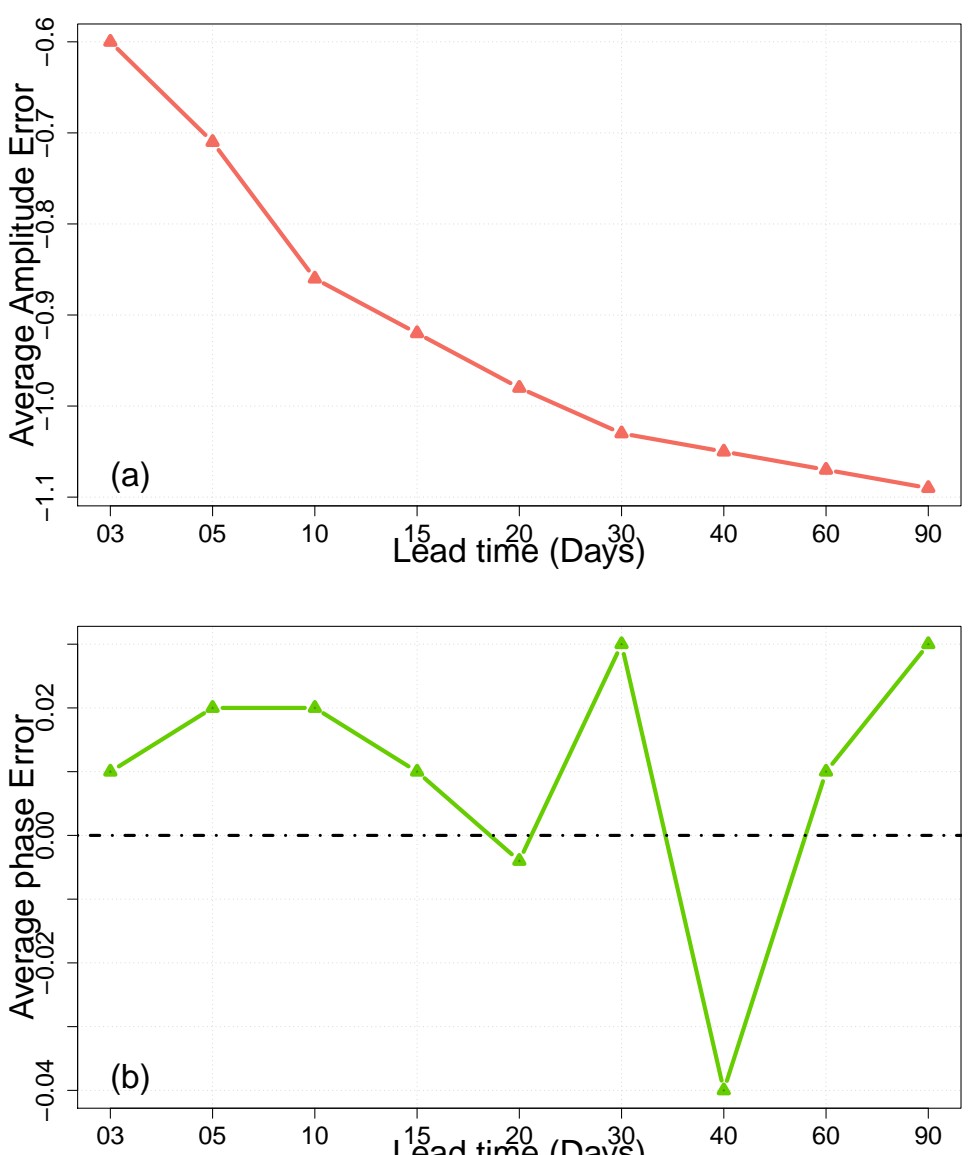

**Figure 11.** MJO Amplitude and phase average errors over all seasons for the period from 1979 to 2020. We notice that the amplitude is underestimated and the phase is well predicted by comparing predictions to forecast.

## 6 Comparison of the SWG forecast with other forecasts

The assessment of the forecast of MJO amplitude with SWG and analogs of Z500 shows good skill until 40 days using probabilistic scores (CRPSS vs. climatology is 0.2 and CRPSS vs. Persistence is 0.4) and scalar scores ($COR = 0.54$ and $RMSE = 1.30$) as explained in sections 5.1 and 5.2. The SWG forecast shows a positive improvement compared to the climatology and the persistence within 40 days Figure 6. In addition, the ROC curve confirmed the ability of the SWG forecast to distinguish between the active and inactive MJO amplitude as shown in Figure 7. The same result was obtained using the ensemble mean of RMM1 and RMM2 as represented in Figure 8. The SWG forecast of RMM1 and RMM2 has good skill within 30 – 40 days respecting the threshold of 0.5 for the COR and $\sqrt{2}$ for RMSE. The difference on the lead time of the forecast depends on the seasons as represented in figure 10. This is consistent with Wu et al. (2016a); Wheeler and Weickmann (2001); Rashid et al. (2011), who found significant differences of skill scores between seasons. We find that the forecast has skill until 30 days for DJF and MAM (with $RMSE = \sqrt{2}$) and 40 days for JJA and SON (with $COR = 0.5$) as shown in Figure 10. This is different from Rashid et al. (2011) and Silini et al. (2021) who obtain higher forecast skill in the winter. However, it is consistent with the results of Wu et al. (2016b) and Vitart (2017) who found higher skill scores for JJA.

We assessed the forecast skill of the SWG with other forecast. We selected two models POAMA (the Australian Bureau of Meteorology coupled ocean-atmosphere seasonal prediction system) and ECMWF, which are providing respectively probabilistic and deterministic forecast of the MJO. We compared mainly the maximum lead time of the MJO amplitude forecast. The POAMA model provides 10-member ensemble. In hindcast mode, the POAMA model has skill up to 21 days (Rashid et al., 2011). The ECMWF reforecasts with Cycle 46r1 has skill to around 40 days. For the error of the amplitude and phase, we found that ECMWF reforecast shows lower amplitude and phase averaged errors compared to those from the SWG forecasts. However, what we found is consistent with other dynamical models (Kim et al., 2018) where they overestimate or underestimate the amplitude and the phase of the MJO.

In addition, we compared quantitatively the SWG forecast with the ECMWF forecast Figure 12. The ECMWF reforecats were taken from (Silini et al., 2022). We found that the ECMWF forecast has highest correlation until 20 days compared to the SWG forecast. The RMSE values of ECMWF forecast are always small compared to the SWG forecast, which indicates a good reliability skill of the ECMWF forecast for lead times of 5 and 10 days. However, for lead times of 20 days the RMSE of ECMWF forecast coincides with the RMSE of the SWG, which shows the improvement of the SWG forecast to lead time above 20 days. The skill scores of the ECMWF (COR and RMSE) (Silini et al., 2022) are computed for each lead time, which is different from our way of computing the skill score (considering the average lead time). Of course, this comparison was made to check the performance of our forecast and not to say that the SWG model can replace a numerical prediction.

We also compared the SWG forecast skill with a machine learning forecast of MJO indices (RMM1 and RMM2) (Silini et al., 2021) . Silini et al. (2021) explored the skill forecast of two artificial neural network types, FFNN (feed-forward neural network) and the AR-RNN (autoregressive recurrent neural network) on MJO indices. Silini et al. (2021) found that the machine learning method gives good skill scores until 26 – 27 days with respect to the standard thresholds of COR and RMSE. We compared the skill scores (RMSE and COR) of the SWG and Silini et al. (2021) forecasts for all lead times. We found that the two models

have the same correlation until 10 days. After 10 days, the correlation of Silini et al. (2021) forecasts decreases rapidly while the correlation of SWG is still significant. For the RMSE, we found that the SWG has smaller values for lead time of 10 days. This indicates that the SWG forecast is more reliable. However, from 30 days, the RMSE of the two models starts to be the same.

To sum up, the comparison of SWG forecast to ECMWF and Silini et al. (2021) forecasts shows that for small lead time (up to 10 days) the ECMWF forecast has better skill. However, the SWG shows positive improvement for long lead times.

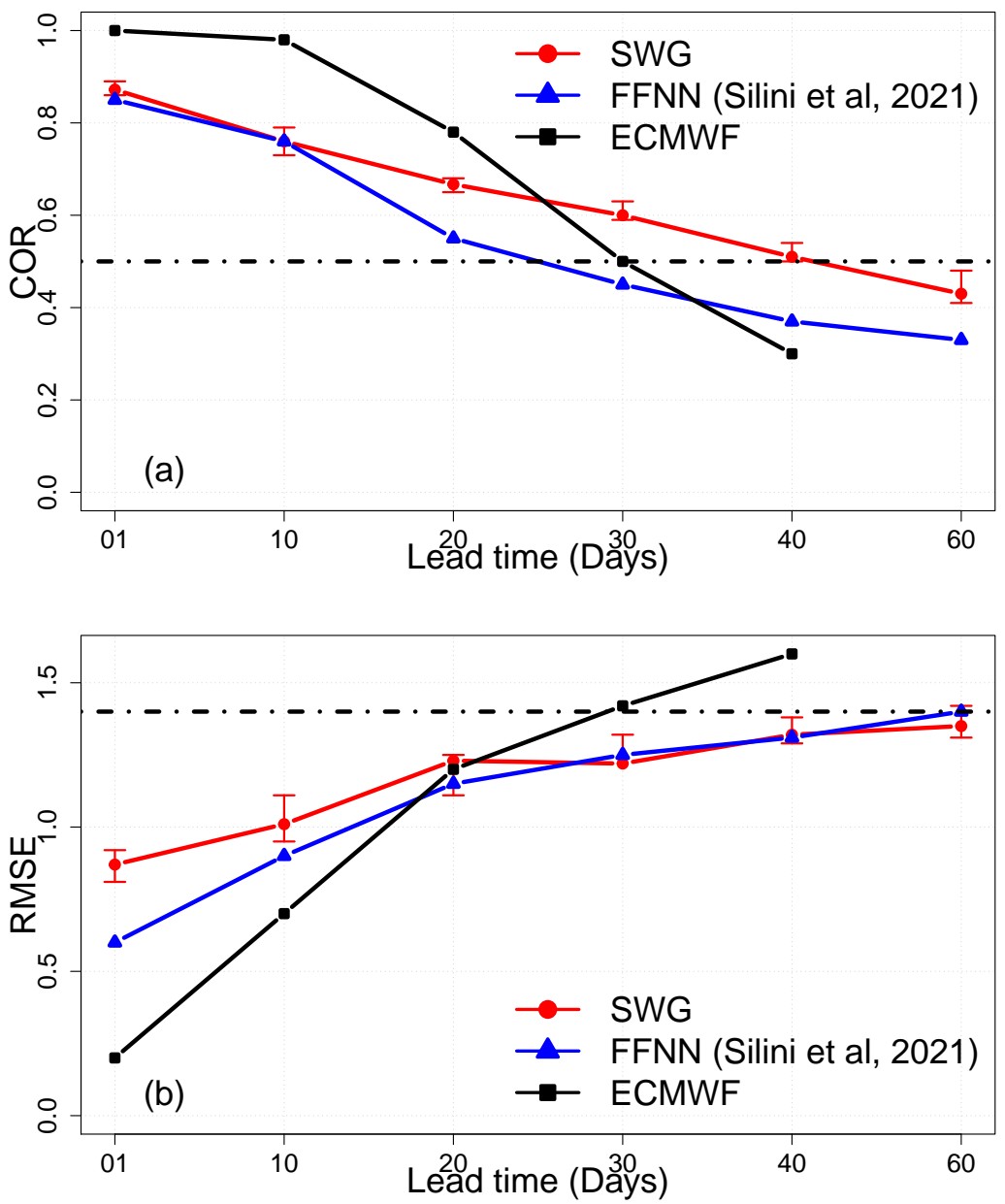

**Figure 12.** Comparison of the values of COR (a) and RMSE (b) between the SWG forecast and forecasts of Silini et al. (2021) (blue lines) and ECMWF (black lines). Confidence intervals for SWG (red lines) were obtained by with a bootstrap procedure over the simulations (1000 samples).

## 7 Conclusions

We performed an ensemble forecast of the MJO amplitude using analogs of the atmospheric circulation and a stochastic weather generator. We used the Z500 as a driver of the circulation Figure 4 over the Indian ocean Figure 2 and we considered analogs from the same phase to provide the forecast for the subseasonal lead time. We explored two ways to forecast the MJO, starting by forecasting directly the daily amplitude, then the daily MJO indices, RMM1 and RMM2, from 1979 to 2020.

We assessed the forecast skill of the MJO forecast by evaluating the ensemble member and the mean of the ensemble member, using respectively probabilistic and scalar verification methods. This allowed us to evaluate the forecast and also to explore the difference between the two verification methods.

We used probabilistic skill scores as the CRPSS and the AUC of the ROC curve Table 1. We found that the forecast showed positive improvement over the persistence and the climatology within 40 days (CRPSS Figure 6). The SWG forecast of the MJO amplitude also showed the capacity to distinguish between active and inactive MJO (ROC curve Figure 7) for the different lead times. Using the scalar scores (COR and RMSE) and the ensemble mean of the forecast of RMM1 and RMM2, we found that the SWG is able to forecast the MJO indices (RMM1 and RMM2) within 30 – 40 days.

We found that the forecast is sensitive to seasons Figure 10. The forecast has skill up to 30 days for the boreal winter (DJF and MAM), while it goes to 40 days for the boreal summer (JJA) and SON. That was consistent with previous studies (Silini et al., 2021; Rashid et al., 2011; Vitart et al., 2017). We also noticed that the forecast of the phase is better than of the amplitude according to the errors for amplitude and phase Figure 11. Finally, we found that the SWG had improvement over the ECMWF forecast for long lead times ($T > 30$ days) and a machine learning forecast (Silini et al., 2021) forecasts for lead times $T > 20 days$.

This paper hence confirms the skill of the SWG to generate ensembles of MJO indices forecasts from analogs of circulation. Such information would be useful to forecast impact variables such as precipitation and temperature.

*Code availability.* The code and data files are available at http://doi.org/10.5281/zenodo.4524562

*Acknowledgements.* This work is part of the EU International Training Network (ITN) Climate Advanced Forecasting of subseasonal Extremes (CAFE). The project receives funding from the European Union's Horizon 2020 research and innovation programme under the Marie Skłodowska-Curie Grant Agreement No 813844. Authors would like to thank A. Corral and M. Minjares for the discussions.

*Author contributions.* MK designed, performed the analyses and wrote the manuscript. PY co-designed the analyses. RS provided data for comparison.

## Appendix A: Comparison of the forecast skill of the MJO using analogs computed from Z500, Z300 and OLR fields

We did the forecast of RMM1 and RMM2 using analogs of Z300 (Figure A4), OLR (Figure A3) and the zonal wind at 250 hPa and 850 hPa (figure 4). The aim of using different atmospheric fields to compute analogs, is to choose the analogs circulation for the MJO forcast with the SWG as explained previously in section 4. We found that the SWG based on analogs of Z300 yields good skills ($COR > 0.5$ and $RMSE < \sqrt{2}$) within $T = 60$ days (Figure 4). However, the skill of the forecast is better for small lead times $\leq 30$ days with analogs of Z500 (Figure 4). We checked the sensitivity of the forecast to seasons as illustrated in Figures A2 and A1 using separately analogs of Z500 and Z300. We compared the COR and the RMSE for different lead times Figures A2 and A1. We found that the RMSE values for the SWG forecast based on analogs of Z300 are the same as the forecast from analogs of Z500 for the different seasons and at the different lead times Figure A2. The RMSE for SON and JJA is lower than the threshold for the $T$ from 3 to 90 days for both forecast Figure A2. However, for DJF and MAM the SWG forecast reaches the threshold of $\sqrt{2}$ respectively at 37 days with analogs of Z300 which is slightly higher than the maximum lead time with Z500 Figure A2. The COR is slightly higher with analogs of Z500 at different lead times Figure A1. However, the threshold of 0.5 is exceeded with forecast based on analogs of Z300 Figure A1.

In this part, we also show the time series for the forecast at different lead times $T = 3, 5, 10$ days for the same year 1986 for the SWG forecast with analogs circulation computed from OLR Figure A3 and from Z300 hPa Figure A4. We notice that the correlation between the mean of the simulations (red line) and the observations of the MJO amplitude are better correlated with SWG forecast based on analogs of Z300 Figure A4. than the one based on analogs of OLR Figure A3.

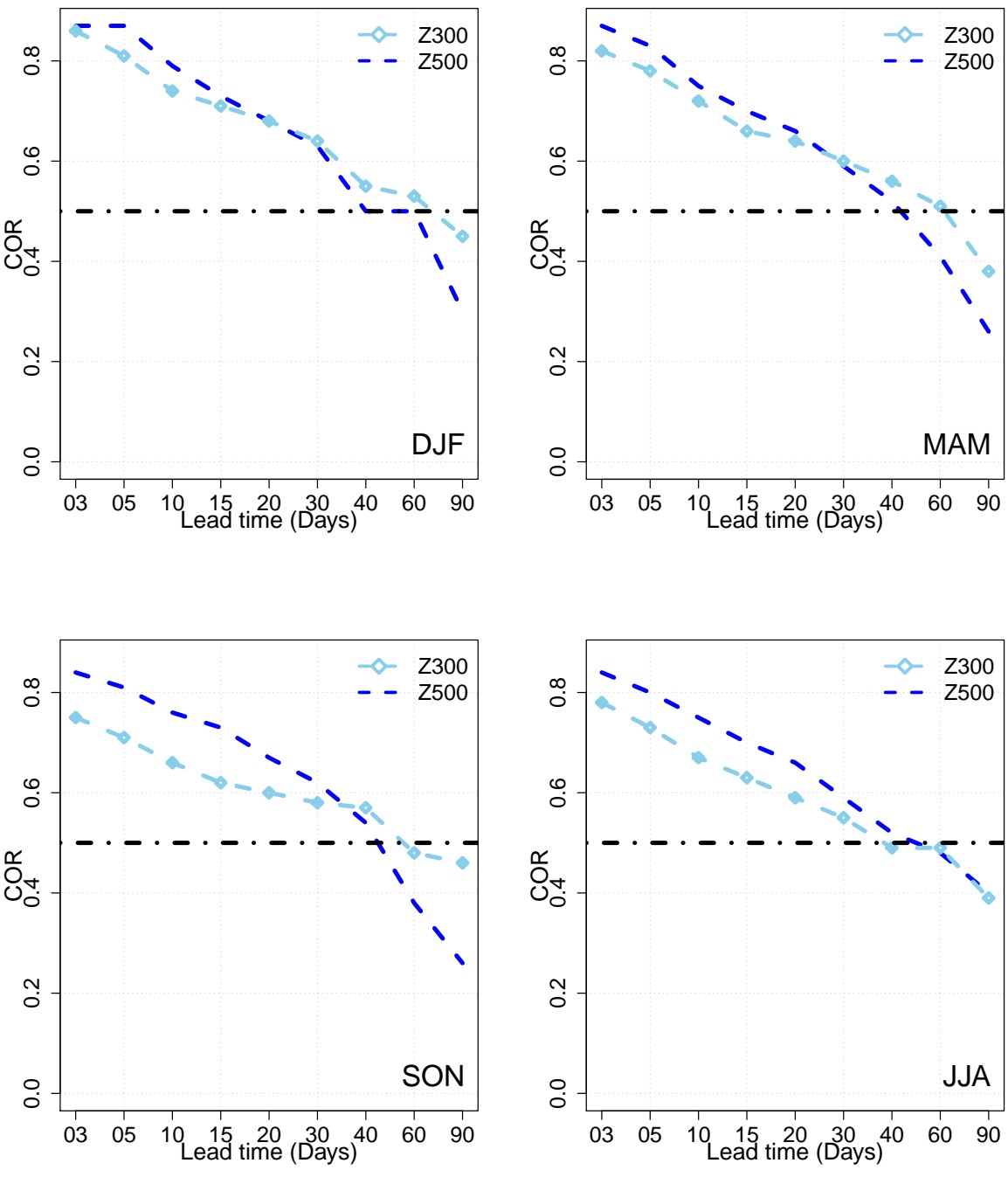

**Figure A1.** COR values for different lead times of forecast from 3 days to 90 days over the period from 1979 to 2020 for the SWG forecast based on analogs of Z500 and Z300 for different seasons (DJF, JJA, MAM and SON).

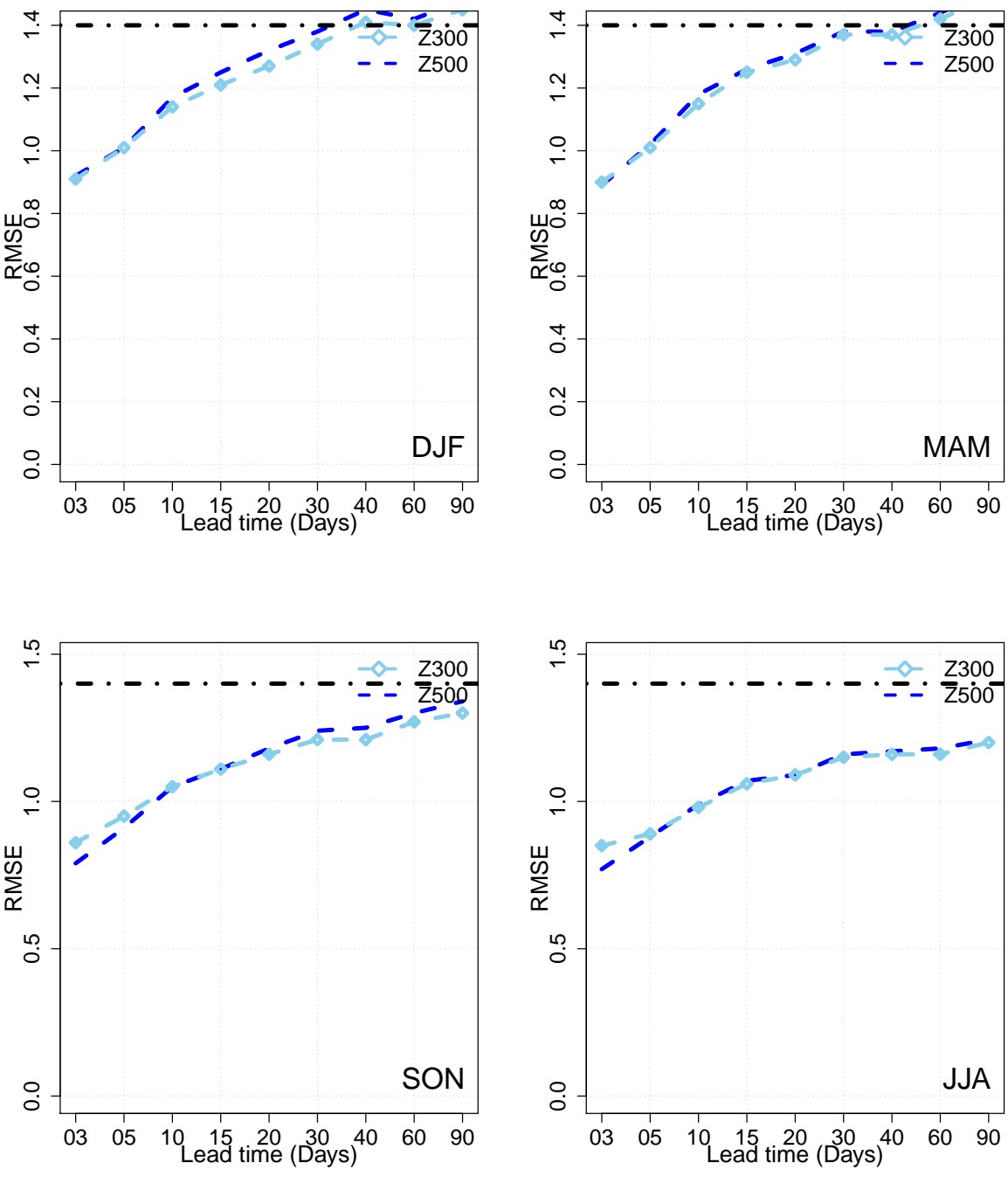

**Figure A2.** RMSE values for different lead times of forecast from 3 days to 90 days over the period from 1979 to 2020 for the SWG forecast based on analogs of Z500 and Z300 for different seasons (DJF, JJA, MAM and SON).

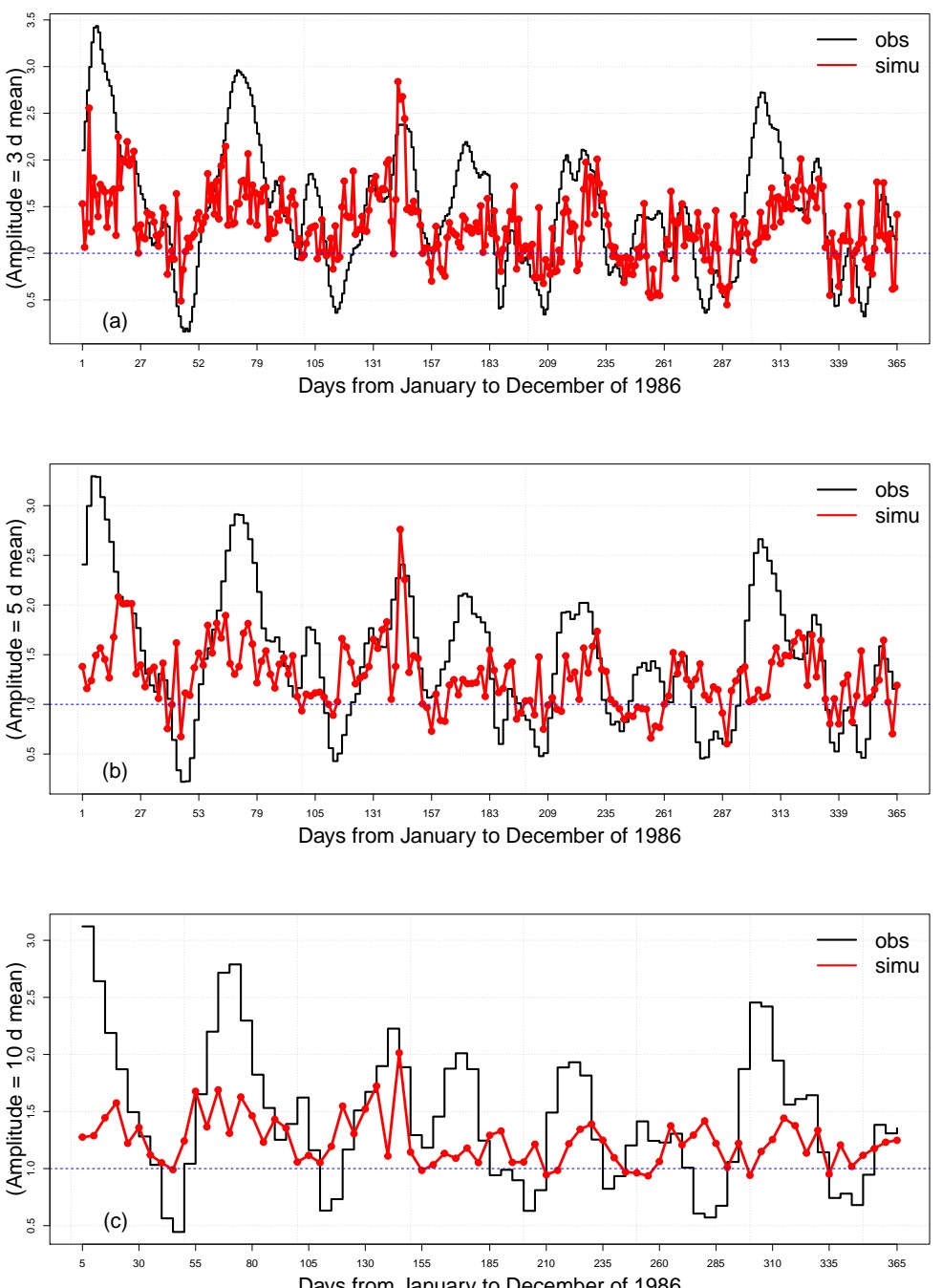

**Figure A3.** Time series of observations and simulations of the MJO Amplitude computed from analogs of OLR, for lead time of 3, 5 and 10 days, respectively (a), (b) and (c), for the year 1986. The red line represents the mean of the 100 simulations, the black line represents the observations, and the blue line indicates the threshold of the MJO activity $A(t) > 1$.

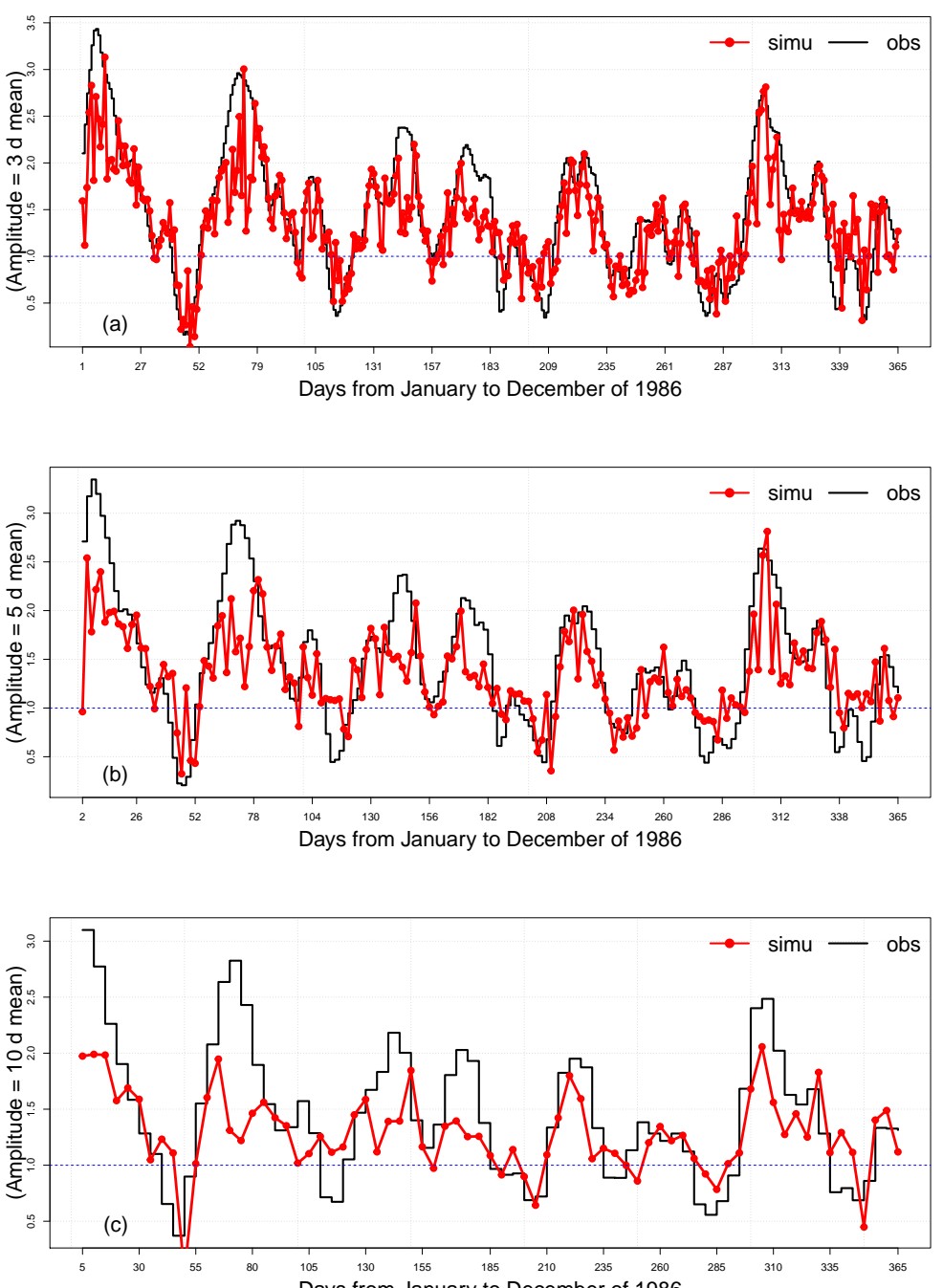

**Figure A4.** Time series of observations and simulations of the MJO Amplitude computed from analogs of Z300, for lead time of 3, 5 and 10 days, respectively (a), (b) and (c), for the year 1986. The red line represents the mean of the 100 simulations, the black line represents the observations, and the blue line indicates the threshold of the MJO activity $A(t) > 1$.

## Appendix B: Domains of computation of analogs

We show in the figure B2 the bivariate correlation (COR) and the RMSE from different geographical regions that we represent in figure B1. The different geographical regions shown in Figure B2 were used to adjust the geographical region to compute analogs.

The COR reaches the threshold of 0.5 at $T = 40$ days for the geographical region with coordinate (50°E – 85°E; 15°S – 15°N) Figure B2. The same result is found for the region with coordinates (60°E – 120°E; 15°S – 15°N) Figure B2 (light blue line). However, the COR is lower for the other lead times $T = 3, 10, 20, 30$ days compared to the one for the region (50°E – 85°E; 15°S – 15°N). For the region with the coordinates (85°E – 120°E; 15°S – 15°N), the threshold of 0.5 for the COR is obtained at a lead time of 34 days Figure B2 (green line). For the region with coordinates (90°E – 150°E; 15°S – 15°N), the forecast skill is significant with COR 0.5, at $T = 30$ days Figure B2 (orange line), which remains the same results for this region compared to (Silini et al., 2022). The RMSE for the different regions is quite the same Figure B2, even if the values for the region (50°E – 85°E; 15°S – 15°N) are slightly lower within 30 days. Therefore the skill forecast (using the bivariate correlation and the RMSE) remains higher for the considered geographical region with the coordinates (50°E – 85°E; 15°S – 15°N).

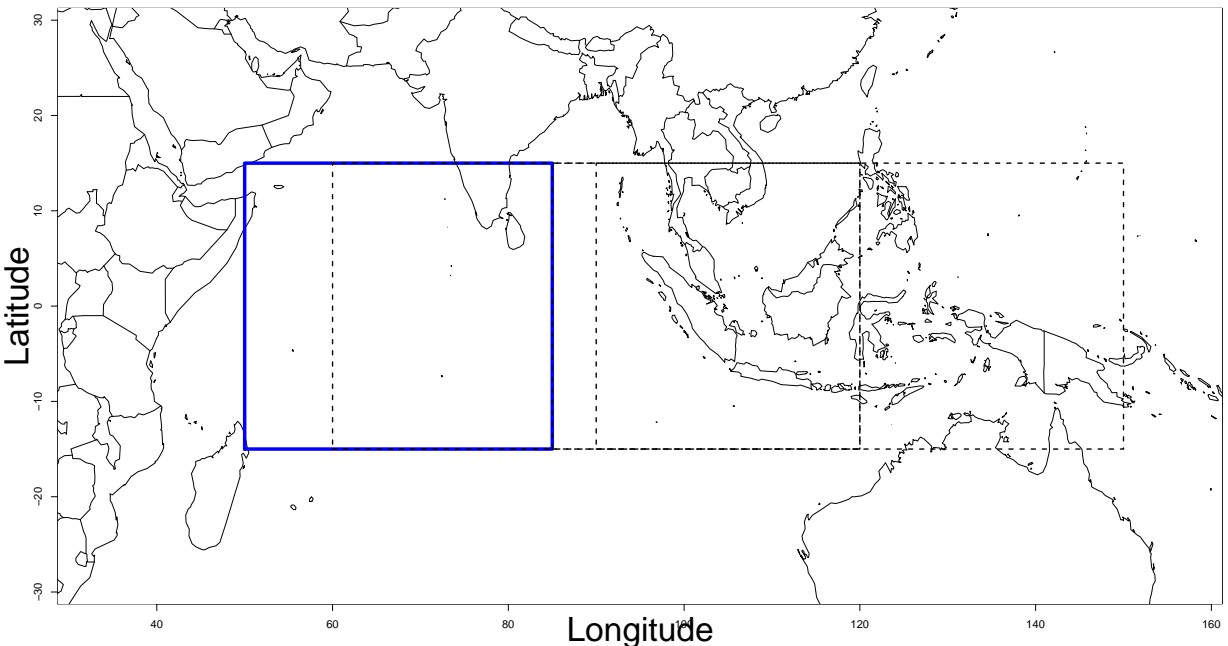

**Figure B1.** Domains of computation of analogs, we computed analogs over the Indian ocean with coordinates (50°E, 85°E – 15°S,15°N) blue rectangle, the Indian-Pacific ocean with coordinates (85°E , 120°E – 15°S , 15°N) and the Indian-maritime ocean with coordinates (90°E , 150°E – 15°S , 15°N).

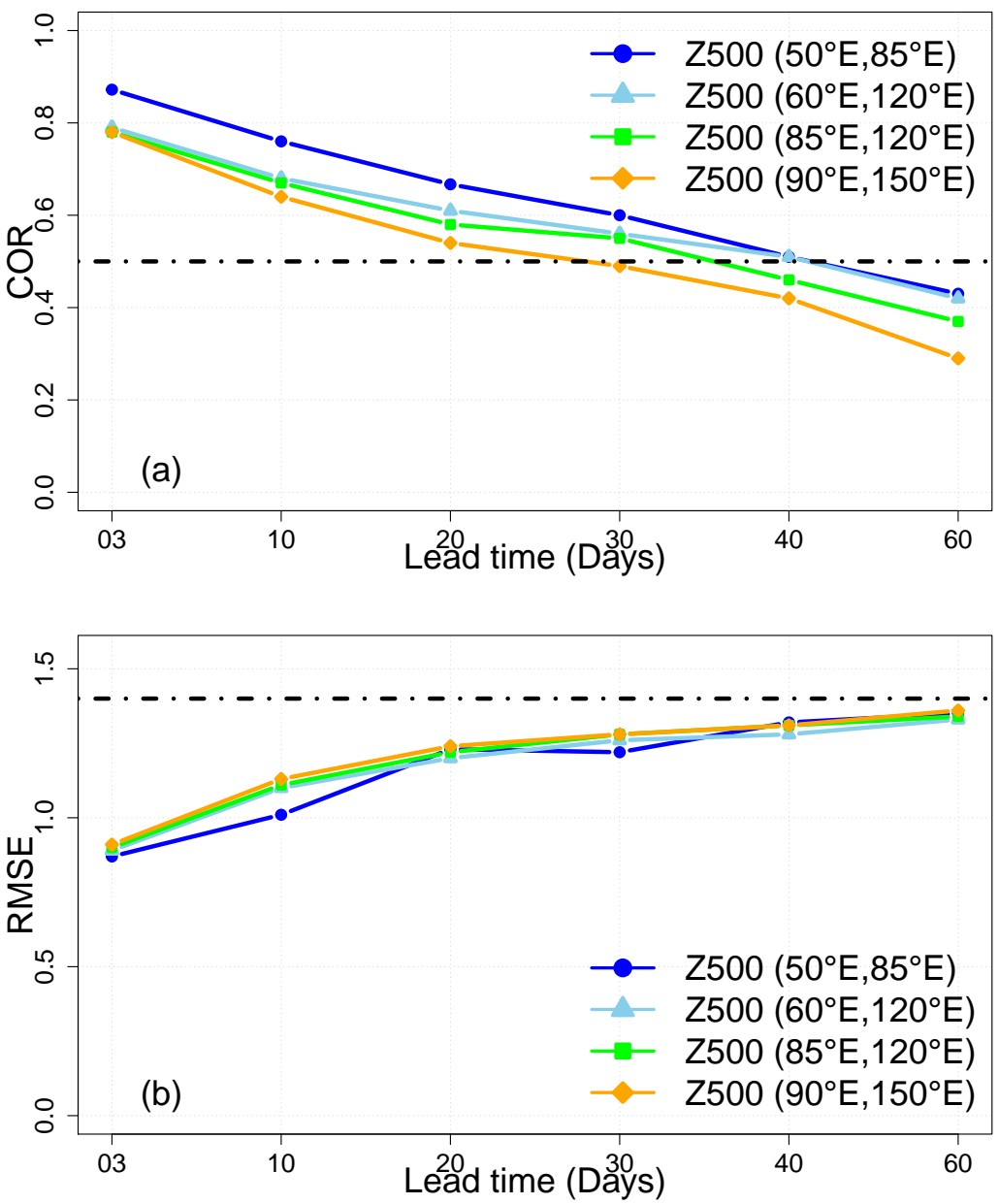

**Figure B2.** Comparison between the COR and RMSE, respectively (a) and (b), of the SWG forecast based on analogs of Z500 computed over different geographical regions, for lead times going from 3 to 60 days over the period from 1979 to 2020. The forecast was made with analogs computed over the Indian ocean with coordinates (50°E , 85°E – 15°S , 15°N) and (60°E , 120°E – 15°S , 15°N), the Indian-Pacific ocean with coordinates (85°E , 120°E – 15°S , 15°N) and the Indian-maritime ocean with coordinates (90°E , 150°E – 15°S , 15°N). As the latitude is the same for the different considered geographical regions, we just mentioned the longitude of each domain in the legend.

## Appendix C: Dependence of the forecast skill on MJO phases

We checked the dependence of the SWG forecast skill of the amplitude of the MJO and the MJO phases. We verified the relationship between the CRPS at $T = 5$ days and the MJO phases Figure C1. We divided the CRPS values in two classes:

– CRPS values above the $75^{th}$ quantile C1 (a),

– CRPS values below the $25^{th}$ quantile C1 (b).

As shown in Figure C1 the difference between the boxplots in the two cases is smaller. Hence, we can say that the dependence of the forecast skill of the MJO amplitude with SWG and the MJO phases is small.

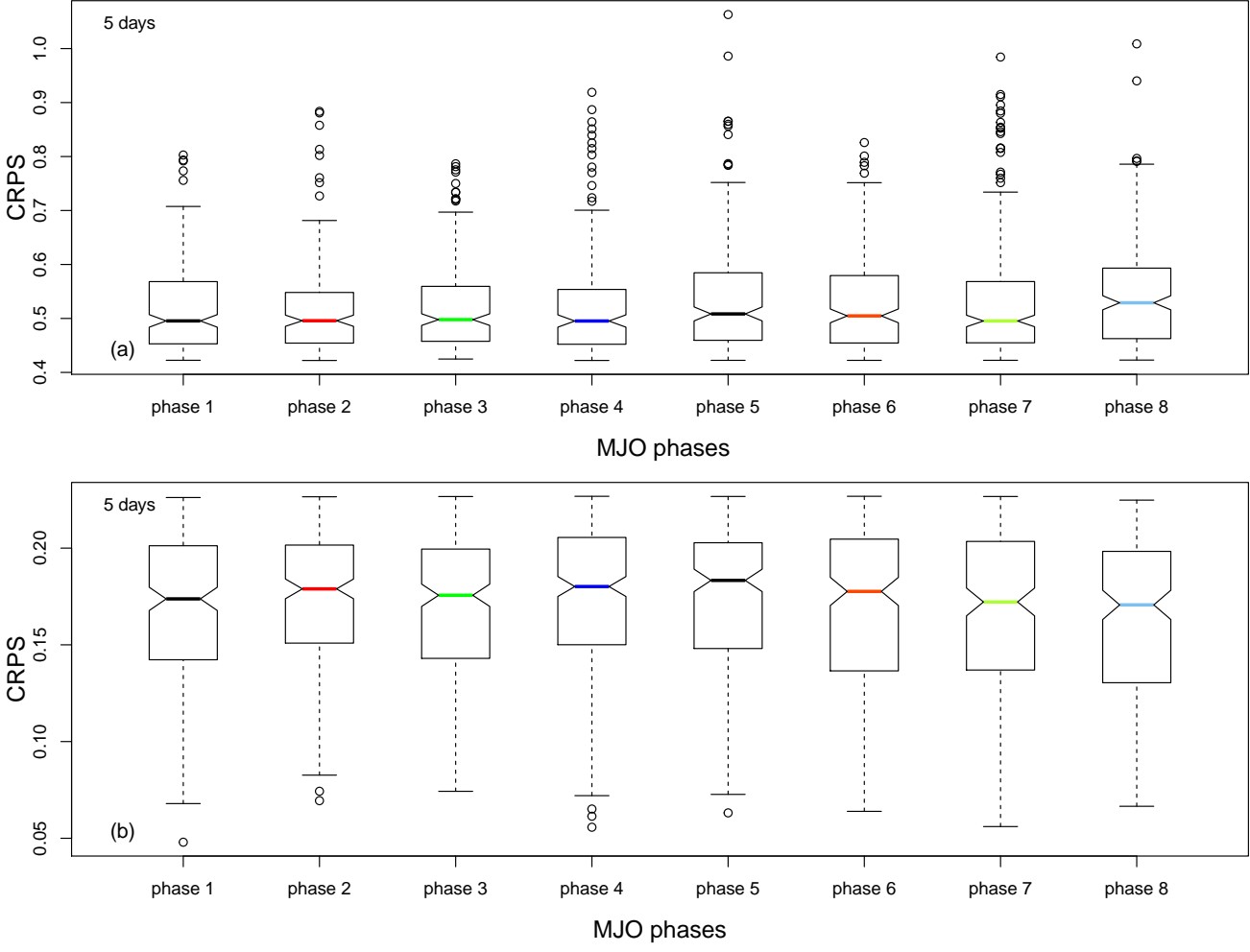

**Figure C1.** Relationship between CRPS and MJO phases. (a) CRPS values above the 75th quantile and (b) CRPS values above the 25th quantile.

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
