# Peer review of "Ensemble forecast of an index of the Madden Julian Oscillation using a stochastic weather generator based on circulation analogs"

_EGUsphere, 2022_

## Author Comment (AC1)

**We thank the reviewer for taking the time to read our manuscript and for the comments. This document is a point-by-point reply to the comments.**

General comments:

In this study, the authors presented the performance of ensemble MJO forecasts using a stochastic weather generator based on circulation analogs. As the MJO is an important source of predictability on the subseasonal time scale, a useful forecast of the MJO is of significant scientific and practical values. Although there have been quite a few studies on MJO forecasts, this study uses a unique approach which is novel in this area. The result is interesting. It shows that a useful skill of the MJO can be achieved at a lead-time of 40 days, which is considerably longer than most dynamical and statistical models. The paper is in general clearly written, although some clarifications and edits are needed. A little more reasoning for choice of variables and region for the analogs and explanation of the results would improve the paper.

Specific comments

The MJO is a planetary-scale tropical disturbance, but the tropical region for the analog calculation in the Indian Ocean (Fig. 2) is quite small. It is a little surprising that Z500 in such a small region can provide information for the MJO evolution. On lines 210-214, one reason for the choice is given which is based on the composition of the RMM index. This may explain why OLR is not used, but RMM does not include Z500 either. The MJO has a baroclinic structure, but 500 hPa is in the middle troposphere that cannot capture the vertical structure. In addition, geopotential height in the tropics does not represent well wind fields. Why not using zonal winds at upper or lower troposphere? Some more explanation on the choice of variable, region, and level would be very helpful.

⇒ **We computed analogs from other regions mentioned in figure B1. However, we obtain better results for the forecast scores by focusing on the small area we used. This is explained by the higher quality of analogs (lower distance).**

**As mentioned in our manuscript, we computed analogs with OLR, which is a driver of the MJO. However, the forecast skill with OLR analogs is lower than for Z500 (with analogs over that region). Z300 analogs, which are close to where the MJO takes place, do not yield significant improvement over Z500 analog skill scores.**

**We also computed analogs from the zonal wind at the upper and lower troposphere (250 and 850 hPa). As shown in the figure below (Figure 1), the wind at 250 hPa, 850 hPa, and the ORL do not improve the bivariate correlation and RMSE forecast skill of the MJO index for a longer lead time (above 20 days) over Z500 or Z300, despite the fact they are the driver of the MJO.**

**Therefore, we take in our approach of forecasting RMM indices of the MJO the minimal physical assumptions on how they are constructed and use "all-purpose" predictor fields (Z500 or Z300). It turns out that geopotential height analogs lead to better scores (COR and RMSE) than other variables that are direct predictors of the MJO.**

**One reason for this apparently surprising behavior is that the composites of OLR and wind speed highly depend on the phase of the MJO (see Figure 1 of Kim et al., J. Clim., 2018). As our analog approach is constrained by the choice of a geographical**

region, it misses the spatio-temporal variability of OLR and wind speed during the MJO. Geopotential heights, although less physically and dynamically relevant for the MJO, are more appropriate predictors from the statistical and mathematical constraint of the analog-based method. This will be discussed in the text, although we acknowledge that this is a conjecture of ours.

[Figure]

**Figure 1. COR (a) and RMSE (b) values for lead times of the RMMs forecasts from 3 days to 90 days using analogs of OLR, wind at 250 hPa, wind at 850 hPa, Z300 and Z500.**

Some justification for the choice of region is given on lines 217-219. The dependence of MJO forecast skill on initial phase is in fact not conclusive in previous studies. It would be

interesting to see how this is the case in this study, i.e., the dependence of MJO skill on the initial phase. It would be interesting to see the skill dependence on MJO amplitude as well.

⇒ **The chosen region for the analogs is rather pragmatic, as the spatial correlation of Z500 shows meaningful structures with RMM indices (Figure 2 in the manuscript). We show below (Figure 2) the dependence of the CRPS values when the forecasts are initiated in each of the MJO phases (for this SWG configuration of analogs) and when the lead time is 10d. The differences are fairly small. The paragraph (ll. 217-219) will be rewritten to reflect that our choice is not based on a physical consensus on MJO forecasts, contrary to what we had suggested, but is just coherent with some aspects of the paper of Kim et al. (2018).**

[Figure]

**Figure 2. Relation between CRPS and MJO phases. (a) CRPS values above the 75th quantile and (b) CRPS values above the 25th quantile.**

Section 6: Some introduction is needed for the two hindcasts of numerical models POAMA and ECMWF. More information on model resolution, version, ensemble size, hindcast period, etc., should be provided. A comparison as in Fig. 10 may not be very meaningful when these forecasts are for the different periods.

⇒ **We will provide more descriptions of the numerical models. In Figure 10, the forecasts that we used for the comparison from the ECMWF and machine learning are done over the same period.**

Minor comments:

Line 4: first two principal
⇒ **This will be corrected.**
Line 74: an MJO event
⇒ **This will be corrected.**
Lines 81-82: "over the region covering 15N-15S" is redundant.
⇒ **This will be corrected.**
Figure 2 caption last sentence: It seems the case for RMM2. How about RMM1?
⇒ **This will be corrected.**
Line 228: "other atmospheric circulations" à "other atmospheric variables"
⇒ **This will be rephrased.**
Line 278: the ensemble spread is increasing, instead of decreasing.
⇒ **This will be corrected.**
7b: How is the bias calculated? Is it the average bias of RMM1 and RMM2?
⇒ **The bias is the average bias of RMM1 and RMM2. It is calculated as follows:**

$$bias = Ensemble\ average_{RMM1} - obs_{RMM1}$$

Line 283: remove "the" in front of "a similar"
⇒ **This will be corrected.**
Line 284: A large RMSE does not necessarily mean a large spread.
⇒ **Indeed. This will be mentioned in the text.**

Line 309: Vitart (2017) also found that the MJO skill is higher in JJA for the ECMWF model
⇒ **Thank you for this reference. We will add it in our discussion.**

Line 351: machine learning
⇒ **This will be corrected.**

Reference:

Vitart, F., 2017: Madden-Julian Oscillation prediction and teleconnections in the S2S database. Quarterly Journal of the Royal Meteorological Society, 143, 2210-2220.

---

## Author Comment (AC2)

**We thank the reviewer for taking the time to read our manuscript and for the comments. This document is a point-by-point reply to the comments.**

Summary:

In this study a stochastic weather generator (SWG) based on the model analogs of the atmospheric circulation is formulated to predict the daily MJO index for a subseasonal lead time. The SWG method adopted by this study has been used to forecast climatic variables, precipitation and the North Atlantic Oscillation by the same authors, and this is the first time that they extend this method for the MJO prediction. The performance of the proposed method is compared against persistence, climatology, and state-of-the-art numerical models. In general the proposed method shows superior performance than both persistence and climatology forecast. Comparison against the full GCM model such as ECMWF forecast shows that the proposed model show larger RMSE (lower COR) than ECMWF model forecast for the 20-day forecast, but smaller RMSE (higher COR) than ECMWF model forecast for days 20-60.

In general, this paper is well written, and the method they proposed is interesting. One significant advantage of their method is the substantially low computational cost compared to a full GCM model (since they use the past model outputs to find the analog), but with superior performance than a full GCM model forecast for days 20-60.

It would be preferable if the authors can clarify why different variables are chosen to form the MJO index and the analog. Besides, I have some other comments shown below.

Recommendation: Major revision

Major Comments:

1. Why different variables and areas are used for RMM, and analog calculation?

Around Line 59: RMM1/2 are calculated from the satellite-derived OLR, and zonal wind at 250hPa, and 850hPa.

Around Line 84: analogs are calculated from the modeled geopotential at 500hPa, 300hPa, and OLR daily data.

Based on these, it seems you are using different variables to calculate RMM and analogs.

⇒ **The RMM1 and RMM2 are computed based on the wind at 250, 850 hPa and the ORL. In our study we have not computed the RMMs ourselves: we used RMMs from the IRI repository. However, we computed the analogs from other atmospheric variables (Z500, Z300 and OLR) as mentioned in the manuscript.**

Although we do know how RMM indices are computed, this paper deliberately makes no (or very few) assumptions on their computation, and assesses how an "all purpose" predictor (e.g. Z500 that is not included in the definition of RMM) can predict RMM. In practice, analogs of OLR or zonal wind speed at 850 hPa were tested for the prediction, but the results are not optimal. One of the reasons stems from an examination of Figure 1 by Kim et al. (J. Clim., 31, 23; 10.1175/JCLI-D-18-0210.1, 2018), which shows the large longitudinal extent of OLR and wind speed anomalies. On such a large "window", the analogs yield rather large distances or low correlations. This implies that the analog SWG gets low skill scores, because the analogs are not very informative. The OLR or zonal wind speed analogs were computed on the optimal window obtained for Z500 or Z300 (figure below), and we find that the COR and RMSE scores are lower than for Z300 and Z500. This is because this "small" window is not appropriate for OLR or wind speed, as reflected in Figure 1 by Kim et al. (2018). This is a potential caveat (or feature) of analogs. The analog geometry needs to be imposed a priori in a rather simplistic way, which does not follow the spatio-temporal features of the MJO, which are known independently. This will be discussed in the text.

My questions are:

(1) what is the motivation to use geopotential at 500hPa & 300hPa, instead of zonal wind at 250hPa and 850hPa, to calculate the analog?
⇒ **As stated above, the motivation is based on trial-and-error computation of forecast skill scores with analogs from various fields and geographical window sizes. The analog method does not seem optimal for forecasts with OLR and zonal wind due to the complexity of their spatio temporal structures. This will be mentioned in the discussion section.**

(2) Have you tried to use zonal wind at 250hPa and 850hPa to calculate the analog?
**We considered analogs from the zonal wind at 250 hPa and 850 hPa. The result is shown in the Figure below. The analog of the zonal wind does not help to get better forecast skill (bivariate correlation and the RMSE) than Z500 or Z300 analogs.**

[Figure]

**Figure. COR (a) and RMSE (b) values for lead times of the RMMs forecasts from 3 days to 90 days using analogs of OLR, wind at 250 hPa, wind at 850 hPa, Z300 and Z500.**

(3) For the analog computation, you used the model data from NCEP. Can you indicate which reanalysis datasets you are using? Are you using CFS-R from NCEP?

⇒ **We used NCEP reanalysis data: NCEP-DOE AMIP-II Reanalysis (R-2): M. Kanamitsu, W. Ebisuzaki, J. Woollen, S-K Yang, J.J. Hnilo, M. Fiorino, and G. L. Potter. 1631-1643, Nov 2002, Bulletin of the American Meteorological Society.**

(4) Around line 59: RMM1&2 are calculated over the region between 15 deg N/S, while the analog is calculated based on the Indian ocean (around Line 100). I understand this is because Indian ocean is the onset place where the MJO occurs. My question is: is it necessary to only form the analog only based on the Indian Ocean? It seems that the analog formation process can be easily extended to later regions where MJO occurs. This might help the case where the initial signal is not well-captured by the initial analogs in the Indian Ocean.

⇒ **Indeed we computed analogs for other regions as shown in Figure B1. We tried to check the influence of the regions on the CRPS, but we did not notice any dependence.**

2.It might be better to reorganize the section 3.2 "Configuration of the stochastic weather generator"

⇒ **Ok, we will rephrase this section.**

(1) Around Lines 110-120: Is it possible to plot some schematics to illustrate the SWG process?

⇒ **We will add more explanations.**

(2) Line 110: "The random selection …that are computed are the products of two weights…rules": rephrase the sentence. Also, it would be better to write an explicit equation combining \w_k^c and \w_k^\{Phi}. Though Lines 111-118 mention how the three w terms related, but it is better to reorder the sequence here and show the explicit equations for each w term.

⇒ **We will reformulate and explicit this part further.**

(3) Line 129 "The persistence and climatological forecasts are randomized by adding a small Gaussian noise": Can you further clarify what kind of Gaussian noise did you add? How did you determine the magnitude of the variance of Gaussian noise (Any justification)?

⇒ **We use white Gaussian noise (mean=0, standard deviation = 1).**

Page 30, Figure B1: Why there are so many triangular white zones in the figure? What variable (indicated by color) is plotted in this figure?

⇒ **We will modify this figure. Figure B1 shows the different regions where we computed the analogs.**

Minor Comments:

Line 11: "We compare our SWG forecast with other forecasts of MJO": It might be better if you can give a short summary (1-2 sentences) of the advantages of your methods over other MJO prediction methods in summary

⇒ **We will add a short summary.**

Line 65 "For this paper, …, the RMM1 and RMM2 allow to… (2004)": rephrase the sentence.

⇒ **We will rephrase this sentence.**

Line 74 "For instance, we consider that there is a MJO event when A(t)>=1". Why the value 1 is selected (any references)?

⇒ **Indeed the MJO is declared active when the amplitude A(t) is above or equal to one.**

**The reference is:**

**Lafleur, D. M., Barrett, B. S., & Henderson, G. R. (2015). Some Climatological Aspects of the Madden–Julian Oscillation (MJO), Journal of Climate, 28(15), 6039-6053. https://doi.org/10.1175/JCLI-D-14-00744.1.**

Figure 2: Is it possible to overlay the contour with p-values 0.05 so I know the correlated areas outside your boxed area.

⇒ **We will modify the plot.**

---

## Author Response (AR1)

**We thank the reviewers for taking the time to read our manuscript and for the constructive comments. This document is a point-by-point reply to their comments.**

**Reviewer 1.**

General comments:

In this study, the authors presented the performance of ensemble MJO forecasts using a stochastic weather generator based on circulation analogs. As the MJO is an important source of predictability on the subseasonal time scale, a useful forecast of the MJO is of significant scientific and practical values. Although there have been quite a few studies on MJO forecasts, this study uses a unique approach which is novel in this area. The result is interesting. It shows that a useful skill of the MJO can be achieved at a lead-time of 40 days, which is considerably longer than most dynamical and statistical models. The paper is in general clearly written, although some clarifications and edits are needed. A little more reasoning for choice of variables and region for the analogs and explanation of the results would improve the paper.

Specific comments

The MJO is a planetary-scale tropical disturbance, but the tropical region for the analog calculation in the Indian Ocean (Fig. 2) is quite small. It is a little surprising that Z500 in such a small region can provide information for the MJO evolution. On lines 210-214, one reason for the choice is given which is based on the composition of the RMM index. This may explain why OLR is not used, but RMM does not include Z500 either. The MJO has a baroclinic structure, but 500 hPa is in the middle troposphere that cannot capture the vertical structure. In addition, geopotential height in the tropics does not represent well wind fields. Why not using zonal winds at upper or lower troposphere? Some more explanation on the choice of variable, region, and level would be very helpful.

⇒ **We computed analogs from other regions as shown in figure B1. However, we obtain better results for the forecast scores by focusing on the small area we used. This is explained by the higher quality of analogs (lower distance).**

**As mentioned in our manuscript, we computed analogs with OLR, which is one of the of the MJO. However, the forecast skill with OLR analogs is lower than for Z500 (with analogs over that region). Z300 analogs, which are close to where the MJO takes place, do not yield significant improvement over Z500 analog skill scores.**

**We also computed analogs from the zonal wind at the upper and lower troposphere (250 and 850 hPa). As shown in the figure below (Figure 1), the wind at 250 hPa, 850 hPa, and the ORL do not improve the bivariate correlation and RMSE forecast skill of the MJO index for a longer lead time (above 20 days) over Z500 or Z300, despite the fact they are drivers of the MJO. One reason for this apparently surprising behavior is that the composites of OLR and wind speed highly depend on the phase of the MJO (see Figure 1 of Kim et al., J. Clim., 2018). As our analog approach is constrained by choice of a geographical region, it misses the spatio-temporal variability of OLR and wind speed during the MJO.**

**Therefore, It turns out that geopotential height analogs lead to better scores (COR and RMSE) than other variables that are direct predictors of the MJO. This can be explained by the theories of the MJO discussed by Zhang et al. (Rev. Geophys., 2020).**

**The convection and the precipitation in those theories of the MJO are related to moisture and thresholds of geopotential height.**
**This is discussed in the text in section 5.**

[Figure]

**Figure 1. COR (a) and RMSE (b) values for lead times of the RMMs forecasts from 3 days to 90 days using analogs of OLR, wind at 250 hPa, wind at 850 hPa, Z300 and Z500.**

Some justification for the choice of region is given on lines 217-219. The dependence of MJO forecast skill on initial phase is in fact not conclusive in previous studies. It would be interesting to see how this is the case in this study, i.e., the dependence of MJO skill on the initial phase. It would be interesting to see the skill dependence on MJO amplitude as well.

⇒ **The chosen region for the analogs is rather pragmatic, as the spatial correlation of Z500 shows meaningful structures with RMM indices (Figure 2 in the manuscript). We show below (Figure 2) the dependence of the CRPS values when the forecasts are initiated in each of the MJO phases (for this SWG configuration of analogs) and when the lead time is 10 days. The differences are fairly small. We added this result to appendix B.**

[Figure]

Figure 2. Relation between CRPS and MJO phases. (a) CRPS values above the 75th quantile and (b) CRPS values above the 25th quantile.

Section 6: Some introduction is needed for the two hindcasts of numerical models POAMA and ECMWF. More information on model resolution, version, ensemble size, hindcast period, etc., should be provided. A comparison as in Fig. 10 may not be very meaningful when these forecasts are for the different periods.

⇒ **Descriptions of the numerical models are provided in lines 336 to 339. In Figure 10, the forecasts that we used for the comparison from the ECMWF and machine learning are done over the same period.**

Minor comments:

Line 4: first two principal
⇒ **This was corrected.**
Line 74: an MJO event
⇒ **This was corrected.**
Lines 81-82: "over the region covering 15N-15S" is redundant.

⇒ **This was corrected.**

Figure 2 caption last sentence: It seems the case for RMM2. How about RMM1?

⇒ **The last sentence is applied for both RMM1 and RMM2. Indeed, the correlation is small in both cases.**

Line 228: "other atmospheric circulations" à "other atmospheric variables"

⇒ **This was rephrased.**

Line 278: the ensemble spread is increasing, instead of decreasing.

⇒ **This was corrected.**

7b: How is the bias calculated? Is it the average bias of RMM1 and RMM2?

⇒ **The bias is the average bias of RMM1 and RMM2. It is calculated as follows:**

$$bias \;=\; Ensemble\ average_{RMM1} - obs_{RMM1}$$

**The definition was mentioned on the text in lines 302 - 303.**

Line 283: remove "the" in front of "a similar"

⇒ **This was corrected.**

Line 284: A large RMSE does not necessarily mean a large spread.

⇒ **This was mentioned in the text.**

Line 309: Vitart (2017) also found that the MJO skill is higher in JJA for the ECMWF model

⇒ **Thank you for this reference. We added it to our discussion.**

Line 351: machine learning

⇒ **This was corrected.**

Reference:

Vitart, F., 2017: Madden-Julian Oscillation prediction and teleconnections in the S2S database. Quarterly Journal of the Royal Meteorological Society, 143, 2210-2220.

**Reviewr #2**

Summary:

In this study a stochastic weather generator (SWG) based on the model analogs of the atmospheric circulation is formulated to predict the daily MJO index for a subseasonal lead time. The SWG method adopted by this study has been used to forecast climatic variables, precipitation and the North Atlantic Oscillation by the same authors, and this is the first time that they extend this method for the MJO prediction. The performance of the proposed method is compared against persistence, climatology, and state-of-the-art numerical models. In general the proposed method shows superior performance than both persistence and climatology forecast. Comparison against the full GCM model such as ECMWF forecast shows that the proposed model show larger RMSE (lower COR) than ECMWF model forecast for the 20-day forecast, but smaller RMSE (higher COR) than ECMWF model forecast for days 20-60.

In general, this paper is well written, and the method they proposed is interesting. One significant advantage of their method is the substantially low computational cost compared to a full GCM model (since they use the past model outputs to find the analog), but with superior performance than a full GCM model forecast for days 20-60.

It would be preferable if the authors can clarify why different variables are chosen to form the MJO index and the analog. Besides, I have some other comments shown below.

Recommendation: Major revision

Major Comments:

1. Why different variables and areas are used for RMM, and analog calculation?

Around Line 59: RMM1/2 are calculated from the satellite-derived OLR, and zonal wind at 250hPa, and 850hPa.

Around Line 84: analogs are calculated from the modeled geopotential at 500hPa, 300hPa, and OLR daily data.

Based on these, it seems you are using different variables to calculate RMM and analogs.

⇒ **The RMM1 and RMM2 are computed based on the wind at 250, 850 hPa and the ORL. In our study we have not computed the RMMs ourselves: we used RMMs from the IRI repository as mentioned in lines 82 to 84. However, we computed the analogs from other atmospheric variables (Z500, Z300, zonal wind and OLR) as mentioned in the manuscript.**

In this paper, analogs of OLR or zonal wind speed at 850 hPa were tested for the prediction, but the results are not optimal as shown in figure 4 in section 5. One of the reasons stems from an examination of Figure 1 by Kim et al. (J. Clim., 31, 23; 10.1175/JCLI-D-18-0210.1, 2018), which shows the large longitudinal extent of OLR and wind speed anomalies. On such a large "window", the analogs yield rather large distances or low correlations. This implies that the analog SWG gets low skill scores, because the analogs are not very informative. The OLR or zonal wind speed analogs were computed on the optimal window obtained for Z500 or Z300 (figure below), and we find that the COR and RMSE scores are lower than for Z300 and Z500. This is because this "small" window is not appropriate for OLR or wind speed, as reflected in Figure 1 by Kim et al. (2018). This is a potential caveat (or feature) of analogs. The analog geometry needs to be imposed a priori in a rather simplistic way, which does not follow the spatio-temporal features of the MJO, which are known independently. Another explanation would be related to the theories of the MJO as discussed by Zhang et al. (Rev. Geophys., 2020). Indeed, the convection and the precipitation in some theories that explain the MJO are related to moisture and thresholds of geopotential. This was well discussed in the text in section 5.

My questions are:

(1) what is the motivation to use geopotential at 500hPa & 300hPa, instead of zonal wind at 250hPa and 850hPa, to calculate the analog?
⇒ **As stated above, the motivation is based on trial-and-error computation of forecast skill scores with analogs from various fields and geographical window sizes. The analog method does not seem optimal for forecasts with OLR and zonal wind due to the complexity of their spatio temporal structures. We gave more explanations about the use of the Z500 in section 5.**

(2) Have you tried to use zonal wind at 250hPa and 850hPa to calculate the analog?
**We considered analogs from the zonal wind at 250 hPa and 850 hPa. The result is shown in the Figure below. The analog of the zonal wind does not help to get better forecast skill (bivariate correlation and the RMSE) than Z500 or Z300 analogs. We added this to our result in section 5, and we better justified our choice.**

[Figure]

**Figure. COR (a) and RMSE (b) values for lead times of the RMMs forecasts from 3 days to 90 days using analogs of OLR, wind at 250 hPa, wind at 850 hPa, Z300 and Z500.**

(3) For the analog computation, you used the model data from NCEP. Can you indicate which reanalysis datasets you are using? Are you using CFS-R from NCEP?
⇒ **We used NCEP reanalysis data: NCEP- NCAR Reanalysis (R-2): Kistler, R., Kalnay, E., Collins, W., Saha, S., White, G., Woollen, J., Chelliah, M., Ebisuzaki, W., Kanamitsu, M., Kousky, V., van den Dool, H., Jenne, R., and Fiorino, M.: The NCEP-NCAR 50-year reanalysis: Monthly means CD-ROM and documentation,**

**Bulletin of the American Meteorological Society, 82, 247−267, <GotoISI>://000166742900003, 2001.**

(4) Around line 59: RMM1&2 are calculated over the region between 15 deg N/S, while the analog is calculated based on the Indian ocean (around Line 100). I understand this is because Indian ocean is the onset place where the MJO occurs. My question is: is it necessary to only form the analog only based on the Indian Ocean? It seems that the analog formation process can be easily extended to later regions where MJO occurs. This might help the case where the initial signal is not well-captured by the initial analogs in the Indian Ocean.

⇒ **Indeed we computed analogs for other regions as shown in Figure B1. We checked the influence of the regions on the CRPS, but we did not notice any dependence.**

2.It might be better to reorganize the section 3.2 "Configuration of the stochastic weather generator"
⇒ **We rephrased this section.**

(1) Around Lines 110-120: Is it possible to plot some schematics to illustrate the SWG process?
⇒ **We added an illustration with explicit equations to clarify the SWG process (see figure 3 in section 3.2).**

(2) Line 110: "The random selection …that are computed are the products of two weights…rules": rephrase the sentence. Also, it would be better to write an explicit equation combining \w_k^c and \w_k^\{Phi}. Though Lines 111-118 mention how the three w terms related, but it is better to reorder the sequence here and show the explicit equations for each w term.
⇒ **We rephrased this part.**

(3) Line 129 "The persistence and climatological forecasts are randomized by adding a small Gaussian noise": Can you further clarify what kind of Gaussian noise did you add? How did you determine the magnitude of the variance of Gaussian noise (Any justification)?
⇒ **We use white Gaussian noise (mean=0, standard deviation = 0.01).**

Page 30, Figure B1: Why there are so many triangular white zones in the figure? What variable (indicated by color) is plotted in this figure?
⇒ **We modified this figure. As explained in Appendix B, figure B1 shows the different regions where we computed the analogs.**

Minor Comments:

Line 11: "We compare our SWG forecast with other forecasts of MJO": It might be better if you can give a short summary (1-2 sentences) of the advantages of your methods over other MJO prediction methods in summary

⇒ **We added a sentence in the abstract.**

Line 65 "For this paper, …, the RMM1 and RMM2 allow to… (2004)": rephrase the sentence.

⇒ **We rephrased this sentence.**

Line 74 "For instance, we consider that there is a MJO event when A(t)>=1". Why the value 1 is selected (any references)?

⇒ **Indeed the MJO is declared active when the amplitude A(t) is above or equal to one.**

**The reference is:**

**Lafleur, D. M., Barrett, B. S., & Henderson, G. R. (2015). Some Climatological Aspects of the Madden–Julian Oscillation (MJO), Journal of Climate, 28(15), 6039-6053. https://doi.org/10.1175/JCLI-D-14-00744.1.**

Figure 2: Is it possible to overlay the contour with p-values 0.05 so I know the correlated areas outside your boxed area.

⇒ **the p-values were computed for the area under the dashed rectangle in figure 2.**

---

## Author Response (AR2)

We thank both reviewers for taking the time to read our manuscript and providing constructive comments to improve our paper further. This document is point-by-point answers to their comments (in boldface).

**Reviewer 1**

1. For the analog creation, the authors chose a very small region where the MJO onset occurs. Both Reviewer 1 and I raised concern about this choice. The authors showed three candidate regions they tested in Figure B1, and in Page 12, around Line 250, the authors mentioned the experiments show that their selected region leads to better results. It would be essential to show at least 1 figure about the forecast skills by creating analogs from these regions. Based on the authors' descriptions, they have already calculated those quantities. It will be easy for them to show such a figure.

**We show in the figure below the bivariate correlation and the RMSE from different geographical regions that we present in figure B1 in the paper. COR reaches the threshold of 0.5 at 40 days for the "small region" (50°E, 85°E; 15°S, 15°N). The same result is found for the region with coordinates (60°E,120°E; 15°S, 15°N). However, the COR is still lower for the other lead times compared to the one for the "small region" (50°E, 85°E; 15°S, 15°N). For the region with the coordinates (85°E,120°E; 15°S,15°N), the threshold of 0.5 for the COR is obtained at a lead time of 34 days. For the region with coordinates (90°E,150°E; 15°S,15°N), the forecast skill is significant with COR $\leq 0.5$, at 30 days, which remains the same results for this region compared to Silini et al. (ESD, 2022). The RMSE for the different regions is quite the same, even if the values for the "small region" (50°E, 85°E; 15°S, 15°N) are slightly lower within 30 days. Therefore the skill forecast (using the bivariate correlation and the RMSE) remains higher for the considered "small region".**
**We added the figure to the annexes and the results to the discussion.**

[Figure]

[Figure]

2. For Figure 2, I suggested adding contours of p-values <0.05 in it since the correlation shown in this figure is low. The authors promised to add the contours though they didn't. Please add it in Figure 2.

**We are sorry for forgetting to add that in the previous version of the paper. We added the figures below to the manuscript. Dots indicate p-values ≤ 0.05.**

[Figure]

[Figure]

**Reviewer 2**

I am in general satisfied with the response to my previous comments and the revised paper. I think that it is acceptable for publication after some minor revisions.

1) Line 5: change "wind speed" to "zonal wind"
**Ok, we changed this.**

2) Line 83: provide reference and link for "from the University of Columbia"
**We mentioned the link in the reference.**

3) Lines 85-87: It is mentioned that the data were downloaded from NCEP, but the paper did not clearly say that whether the data are the NCEP reanalysis. Also what OLR data are used and a reference should be given.
**Ok we added the reference for the OLR.**

4) Are your results sensitive to the data used? The NCEP reanalysis data are quite old. Can you get the same results using ERA5?
**In our previous paper (Krouma et al,GMD 2022), we checked the difference between simulations using NCEP and ERA5 reanalysis, and we found similar skill score values. Based on this information, in this paper, we also assumed that there is little difference. The NCEP reanalysis is generally more convenient to use (the high resolution of ERA5 is not necessary for the computation of analogs).**
**We computed the COR and RMSE scores with the ERA5 reanalysis (with analogs of Z500), and found slightly lower skill scores. We suspect that the RMM indices would have to be computed on ERA5 fields for improved skill scores.**
**Therefore we believe that our analysis provides an order of magnitude for the performance for the SWG approach.**

5) Lines 101-102: Rephrase the sentence "We explored…".
**Ok we rephrased this sentence.**